# Recommendations for Spectral Fitting of SO₂ from mini-MAX-DOAS Measurements

Zoë Y. W. Davis[1] and Robert McLaren[2].

[1]Department of Earth and Space Science, York University, Toronto, M3J 1P3, Canada
[2]Department of Chemistry, York University, Toronto, M3J 1P3, Canada

*Correspondence to*: Zoë Davis (zoeywd@yorku.ca)

**Abstract.** Fitting SO₂ dSCDs from MAX-DOAS measurements of scattered sunlight is challenging because actinic light intensity is low in wavelength regions where the SO₂ absorption features are strongest. SO₂ dSCDs were fit with different wavelength windows ($\lambda_{low}$ to $\lambda_{high}$) from ambient measurements with calibration cells of $2.2\times10^{17}$ and $2.2\times10^{16}$ molec cm$^{-2}$
inserted in the light path at different viewing elevation angles using an OceanOptics USB2000 spectrometer in a miniature-MAX-DOAS instrument. SO₂ dSCDs were the least accurate and fit errors were the largest for fitting windows with $\lambda_{low} < 307$ nm or $\lambda_{low} > 312$ nm. The SO₂ dSCDs also exhibited an inverse relationship with the depth of the differential features in the SO₂ absorption cross-section for fitting windows with $\lambda_{low} < 307$ nm. Spectra measured at low viewing elevation angles (i.e., $\alpha = 2°$) exhibited less accurate SO₂ dSCDs for the same fitting windows compared to higher angles. The use of a 400 nm short-
pass filter or a polynomial to account for stray light (the offset function), increased the accuracy of the SO₂ dSCDs for many different fitting windows, decreased fit errors, and decreased the dSCDs' dependence on the depth of the SO₂ differential absorption features. These results suggest that the radiance at shorter wavelengths was increased by stray light. The inaccuracies at lower fitting wavelengths were increased by stray light originating from light with $\lambda > 400$ nm. Deviation of the SO₂ dSCD from the true value depended on the SO₂ concentration for some fitting windows rather than exhibiting a
consistent bias. Uncertainties of the SO₂ dSCD reported by the fit algorithm were >50% less than the true error for many windows, particularly for the measurements without the filter or offset function. For retrievals with the filter or offset function, increasing $\lambda_{high} > 320$ nm tended to decrease the reported fit uncertainty but did not increase the accuracy. Based on the results of this study, a short-pass filter and a fitting window of $307.5 < \lambda < 319$ nm are recommended for retrieval of SO₂ SCDs from mini-MAX-DOAS measurements. If a filter is not available or conflicts with other species to be determined (NO₂, HCHO,
etc.), the offset function should be enabled, and a fit window $307.5 < \lambda < 319$ nm is still recommended.

## 1 Introduction

The Differential Optical Absorption Spectroscopy (DOAS) technique has been used since its introduction by Brewer et al. (1973), Noxon (1975), Perner et al. (1976), and Platt et al. (1979) to measure atmospheric species with narrow-band structures
of absorption in the visible and near UV wavelength region. A major challenge for the successful determination of trace-gas of Slant Column Densities (SCDs) using the DOAS method is the optimization of the retrieval parameters (Platt and Stutz,

2008; Vogel et al., 2013). These parameters depend on the atmospheric composition, measurement conditions, and which DOAS instrument is used. The wavelength range of the retrieval ("fitting window") is a key parameter that depends on the differential absorption features of the trace-gas (Vogel et al., 2013). Retrieval of differential SCDs (dSCDs) of $SO_2$ from Multi-Axis-DOAS (MAX-DOAS) measurements is challenging in a number of ways, including because the $SO_2$ absorption features are strongest in the wavelength region where the intensity of solar light becomes relatively small. There are three major regions of photo-absorption by $SO_2$ in the UV range: the very weak absorption in the A band from 340-390 nm, the moderately strong B band from 260-340 nm, and the strongest C band from 180-240 nm. MAX-DOAS spectroscopy commonly uses the $SO_2$ B band in the near UV, which has absorption peaks of increasing strength with decreasing wavelength (Hermans et al., 2009; Xie et al., 2013). Retrievals of $SO_2$ dSCDs in the 300-325 nm range are complicated by the low intensity of scattered sunlight that results in high photon shot noise (Bobrowski et al., 2010). Actinic flux at the surface level of the earth decreases by several orders of magnitude in the 320-290 nm region due to a steep increase in $O_3$ absorption with decreasing wavelengths (Kreuter and Blumthaler, 2009). $O_3$ differential absorption features can also cause interference in the fit because of the similarity to the $SO_2$ differential absorption features between 315 and 326 nm (Rix et al., 2012). An additional challenge due to the low UV intensity is that stray light in spectrometers can be a significant proportion of the measured signal, causing underestimation of the true optical density (and dSCD) and reducing fit accuracy (Bobrowski et al., 2010; Kreuter and Blumthaler, 2009). Stray light impacts tend to be severe for the widely used compact spectrometers, such as from OceanOptics (Bobrowski et al., 2010). The optimal fitting window for retrieval of $SO_2$ column densities from MAX-DOAS spectra must have a lower wavelength ($\lambda_{low}$) small enough to include strong features of $SO_2$ absorption but large enough to ensure enough solar signal and prevent significant stray light effects. The upper wavelength of the fit range ($\lambda_{high}$) should ensure that the fitting window includes multiple $SO_2$ absorption structures while excluding wavelengths where $SO_2$ absorption features are so weak that degrees of freedom (DOF) are unnecessarily increased, increasing fitting uncertainty. MAX-DOAS fit windows must be relatively narrow compared to direct sun viewing applications because the air mass factors used to convert SCDs to vertical column densities (VCD) differ with wavelength due to scattering (Fioletov et al., 2016). An overly broad fit window also risks the inclusion of strong absorption features from other gases (Vogel et al., 2013) and increased errors due to insufficient correction of the broad-band terms (Marquard et al., 2000; Pukite et al., 2010). However, an overly narrow fit window can lead to cross-correlation between the reference absorption cross-sections (Vogel et al., 2013). Inclusion of upper wavelengths with weak $SO_2$ in the fit can improve the fit results by allowing a better distinction between $SO_2$ absorption features and other fit components (e.g., $O_3$ absorption features, Ring spectrum).

A further complication is that for measurements of very large column densities of $SO_2$ (e.g., from volcanic studies), the optimal wavelength window may be present at higher wavelengths where $SO_2$ absorption features are weaker (Bobrowski et al., 2010). High optical densities below 320 nm from large column densities can cause non-linearities in the relationship between the column density and measured optical density in the fit. This phenomenon occurs for large (actual) optical densities if the cross-section in the fit was not recorded with the same spectrometer as the measurements, which is common, and the instrument's spectral resolution is too low to completely resolve the absorption bands (Kern, 2009; Platt and Stutz, 2008). Large column

densities of $SO_2$ result in optical densities in the B band that can exceed unity, violating the assumption in the standard DOAS retrieval of weak absorption with optical depths of less than ~0.1 (Bobrowski et al., 2010; Bobrowski and Platt, 2007; Fickel and Delgado Granados, 2017; Kern, 2009; Platt and Stutz, 2008). Compact spectrometers typically have an insufficient spectral resolution for the optical density of the $SO_2$ absorption bands to be proportional after convolution for large column densities

(Bobrowski et al., 2010; Platt and Stutz, 2008). Consequently, the true column density can be underestimated because the differential absorption line depths from the standard DOAS convolution approximation can be greater than mathematically correct convolution (Bobrowski et al., 2010; Kern, 2009; Yang et al., 2007). Underestimation has been shown to increase with decreasing wavelength from 320-300 nm and increasing column density of $SO_2$ (Kern, 2009). This effect is important for volcanic plume studies and in the most polluted urban and industrial environments

Despite the importance of using an optimal fitting window, various windows have been used to retrieve MAX-DOAS $SO_2$ SCDs in the literature, and few studies attempted to assess the impact of the window's wavelength range on the $SO_2$ SCDs (Vogel et al., 2013). Fitting windows in previous MAX-DOAS studies include 305-317.5 nm (Tan et al., 2018), 307.5-328 (Schreier et al., 2015), 307.6-325 nm (Jin et al., 2016), 307.8-330 nm (Wang et al., 2017), 310-320 nm (Irie et al., 2011), and 307.5 to 315.0 nm (Bobrowski and Platt, 2007). Salerno et al. (2009) examined the sensitivity of $SO_2$ SCD to the fitting window

in the 300-320 nm region using calibration cells of $SO_2$ of $3.2 \times 10^{17}$ and $8.5 \times 10^{17}$ molec. $cm^{-2}$. An optimal fitting window of 306.7-314.7 nm was determined based on the smallest SCD errors by varying the wavelengths of the fit window. However, the variations of the lower and upper window limits were only conducted for a single fixed upper limit and lower limit, respectively. Also, since the column densities were relatively large, more representative of volcanic plumes, the determined fitting window may not be ideal for smaller $SO_2$ column densities often observed in urban studies. Fickel and Delgado

Granados (2017) observed a high dependence of $SO_2$ SCDs from measurements of a volcano plume on the fitting window, particularly for large column densities. The authors suggested using different fitting windows for different column densities: 310-322 nm for $SO_2$ column densities $<10^{17}$ molec. $cm^{-2}$, 322-334 nm for column densities $>10^{18}$ molec. $cm^{-2}$, and 314.7-326.7 nm for intermediate column densities. A modelling study by Bobrowski et al. (2010) suggested using fitting windows in the higher 360-390 nm range for column densities on the order of $10^{19}$ molec. $cm^{-2}$ because the $SO_2$ absorption features are much

weaker. Therefore, DOAS retrievals of $SO_2$ from instruments that observe a wide range of column densities can switch fitting windows depending on the magnitude of the retrieved SCDs. For example, TROPOMI satellite retrievals of $SO_2$ uses 312-326 nm as a baseline window but uses the alternates 325-335 nm for $SO_2$ SCDs $>4 \times 10^{17}$ molec. $cm^{-2}$ and 360-390 nm for SCDs $> 6.7 \times 10^{18}$ molec. $cm^{-2}$ (Theys et al., 2017).

In this study, MAX-DOAS measurements of two different calibration gas cells with column densities of $SO_2$ representative of

polluted urban conditions were conducted to examine the variation in the retrieved $SO_2$ dSCDs with 1) different fitting windows, 2) different viewing elevation angles ($\alpha$), 3) the use of a 400 nm short-pass filter, and 4) the offset function enabled.

## 2 Methods

The mini-MAX-DOAS instrument (Hoffmann Messtechnik GmbH model #16127) consisted of a sealed metal box with a UV fibre-coupled spectrometer and all electronics inside. Incident scattered sunlight received by the cylindrical black telescope in

front of the entrance optics is focused into the quartz fibre by a cylindrical quartz lens with a focal length of 40 mm. The MAX-DOAS instrument used a relatively low-cost and commonly employed compact spectrometer, an OceanOptics USB2000 spectrograph. The spectrometer has a 50 µm wide entrance slit and a Sony ILX511 linear silicon Charge-Coupled Device (CCD) array detector (2048 pixels, pixel size 14x200 microns, signal-to-noise ratio at full signal 250:1). The spectral range of

the spectrometer is 290-433 nm, with a resolution of ~0.7 nm FWHM in the fitting range used. A Peltier stage cooled the spectrograph to maintain the chosen temperature of $5^{\circ}C$. A stepper motor mounted underneath allows the instrument to point at different α above the horizon. The instrument was connected to a laptop via USB to transfer spectrometer data and allow automated measurements by Jscript programs using the DOASIS software package.

MAX-DOAS spectra of scattered solar light were recorded with an $SO_2$ calibration gas cell (Resonance Ltd.) inserted in the

light path (in the telescope tube). The two cylindrical gas cells with a 22 mm diameter and 14.13 mm thickness had calibrated slant column densities (SCDs) of $2.2\times10^{17}$ molec $cm^{-2}$ (higher) and $2.2\times10^{16}$ (lower) (+/- 10%) molec $cm^{-2}$. Active-DOAS measurements of the $SO_2$ gas cells confirmed the SCDs. These SCDs would be equivalent to an air mass with $SO_2$ mixing ratios of 41 and 4 ppb, respectively, for a $α=30^{\circ}$ measurement within a homogeneous boundary layer of 1 km, assuming the measurement was fit against a zenith pointing reference spectrum observing outside of the $SO_2$-polluted zone, calculated using

Eq. (2) in Davis et al. (2019) For each cell, spectra were recorded around solar noon in September in Toronto, Ontario (43.773 N, -79.506 W) between 12:53 and 13:26 local time at $α = 90^{\circ}, 30^{\circ}, 15^{\circ}, 8^{\circ}, 4^{\circ}$, and $2^{\circ}$ above the horizon, followed by a $90^{\circ}$ measurement without the gas cell. This second zenith measurement was used as the Fraunhofer Reference Spectrum (FRS) in the fit. The time between the $2^{\circ}$ and $90^{\circ}$ measurements in one sequence (both containing the $SO_2$ cell) was less than 13 minutes. The FRS were obtained less than 35 minutes after the beginning of the respective sequence of cell measurements. Each

recorded spectrum was the average of 1000 spectra with an integration time of ~115 ms. The experiment was repeated for both gas cells by placing a 400 nm short-pass filter (Edmund Optics TECHSPEC® OD 2 #47-285) within the telescope between the MAX-DOAS lens and the $SO_2$ gas cell. The fused silica filter had a thickness of 3 mm, a cut-off wavelength of 400 nm, and a transmission wavelength range of 250-385 nm. The blocking optical density was ≥2.0, and the transmission was >85% in the transmission range. Spectra collected using the filter were fit against a FRS collected by measuring a $90^{\circ}$ spectrum

without a gas cell but including the filter.

Trace gas differential Slant Column Densities (dSCDs) were obtained using the DOAS method (Platt and Stutz, 2008) with the DOASIS software (Institute of Environmental Physics, Heidelberg University, 2009). All spectra were corrected for dark current and electronic offset, and wavelength calibrated using measurements of a Mercury (Hg) lamp. Included in all fits were a Fraunhofer Reference Spectrum (FRS), Ring spectrum, a 3rd order polynomial, and absorption cross-sections of $SO_2$ at 293K

and $O_3$ at 293 and 223 K (Bogumil et al., 2003). The shift and squeeze terms were allowed for the fit components with the Ring spectrum terms linked to the FRS terms and the $O_3$ cross-section terms linked to the $SO_2$ cross-section terms (shift limited to -0.2 to 0.2 nm). The shift and squeeze terms are included in DOAS analyses to compensate for wavelength shifts due to instrumental instabilities, such as temperature changes during measurements altering the pixel-to-wavelength calibration

(Lampel et al., 2017; Stutz and Platt, 1996). In the case of the FRS, the shift and squeeze terms also compensate for the "tilt effect" that increases fit residuals by artificially shifting the spectral positions of Fraunhofer and molecular absorptions lines between the measurement and reference spectra that have different viewing elevation angles (Lampel et al., 2017). The tilt effect arises because atmospheric modification of the spectral structures in the spectrum occurs before convolution with the

instrument slit function and the modifications are non-commutative but are applied in the reverse order by the analysis procedure (Lampel et al., 2017). The cross-sections were obtained from the MPI-MAINZ UV/VIS Spectral Atlas of Gaseous Molecules of Atmospheric Interest (Keller-Rudek et al., 2013). The reported uncertainty in the $SO_2$ absorption cross-section is ~3% (Bogumil et al., 2003). DOASIS fits dSCDs using an iterative algorithm based on the Levenberg-Marquardt method that finds the optimal solution by minimizing a cost function. The cost function includes the deviation between the measured

spectrum and the spectrum modelled using the components included in the fit. Details on the DOASIS fitting algorithm can be found in Kraus (2006). The $SO_2$ dSCDs were fit in DOASIS with varying fitting windows using $\lambda_{low}$= 303-318 nm and $\lambda_{high}$= 310-340 nm in ~0.2 nm increments. The "retrieval interval mapping" technique (Vogel et al., 2013) was used to visualized and systematically evaluate the variations in the $SO_2$ dSCDs. The dSCDs are displayed as contour plots where $\lambda_{low}$ and $\lambda_{high}$ are the first and second dimensions, and the dSCDs are denoted using a colour scale.

For each calibration gas cell (higher and lower), four scenarios were fit: i) the base case (B) with no filter and no offset function, ii) no filter with offset function enabled (B+O), iii) with filter and offset disabled (B+F), and iv) with both filter and offset enabled (B+ F+O). $SO_2$ dSCDs were considered "accurate" if within ±10% of the higher calibration cell value and ±50% of the lower calibration cell value, $2.2\times10^{17}$ and $2.2\times10^{16}$ molec cm$^{-2}$, respectively. The background $SO_2$ in the atmosphere in Toronto was assumed to be negligible (<1 ppb) because there are currently no significant sources in Toronto (ECCC, 2018).

A few industrial sources of <1600 tonnes of $SO_2$ yr$^{-1}$ were present south-west of Toronto (ECCC, 2018), but the measurements were conducted under North-Easterly wind conditions. Typical hourly average mixing ratios of $SO_2$ in northern Toronto are <0.5 ppb (Ontario Ministry of the Environment, 2019).

**3 Results**

Examples of spectral retrievals of $SO_2$ from the α=2° spectrum in the base case (no filter and offset function disabled) are

shown in Fig. 1.

**3.1 Higher Concentration Reference Cell**

$SO_2$ dSCDs fit from the α=2° and α=30° measurements using the higher concentration cell are shown in Fig. 2 with varying fitting windows for the four scenarios. The deviations of the $SO_2$ dSCDs from the expected value of $2.2\times10^{17}$ molec cm$^{-2}$ (fit errors) are shown in Fig. 3, where purple and green colours indicate under- and over-estimation, respectively. Grey and black

areas indicate that the dSCD under- and over-estimated the expected value by more than $8\times10^{16}$ molec cm$^{-2}$, respectively. For the base case, the windows with $\lambda_{low}$ <307 nm ("low wavelengths") underestimated the expected $SO_2$ dSCD, as indicated by the grey areas in Fig. 2 (B) and the purple areas in Fig. 3 (B). The addition of the short-pass filter increased the accuracy of the $SO_2$ dSCDs for most windows, especially in the low wavelengths (Figs. 2 & 3 (B+F)). These results suggest that stray light originating from wavelengths >400 nm increased the underestimation of $SO_2$ dSCDs at low wavelengths. Stray light is a well-

known source of interference in spectroscopic measurements that reduces accuracy and can obscure weak spectral lines (Kristensson et al., 2014). Ideally, a spectrometer's detector receives only light with the correct spectral bandwidth window at each pixel (Lindon et al., 2000). Stray light is additional light of an incorrect wavelength that enhances the background signal in ways that can vary across the spectral range (Kristensson et al., 2014). Sources of stray light include imperfections in the diffraction grating, leakage of light into the instrument, and scattering off mirrors and dust inside the instrument (Lindon et al., 2000). Stray light results in apparent negative deviations from Beer's law (Choudhury and Prayagi, 2015), causing an underestimation of the retrieved $SO_2$ dSCD by "filling-in" the measured intensity reduced by $SO_2$ absorption features and an underestimation of the real optical density (Bobrowski et al., 2010). Stray light has an enhanced effect at low wavelengths because of the low measured signal and sensitivity near the lower end of the actinic spectral range (Choudhury and Prayagi, 2015). Many fitting windows with $\lambda_{low}$<307 nm and $\lambda_{high}$< 320 nm still underestimated the $SO_2$ dSCD even with the filter (Fig. 2 (B+F)). The remaining underestimation is likely due to stray light originating from <400 nm and effects of non-linearity between the column density and measured optical density because of the relatively large differential optical densities of $SO_2$ of >0.08 in the regions of strong absorption below 307 nm (Fig. 1) (Kern, 2009; Platt and Stutz, 2008). See Fig. 3.21 in (Kern, 2009) for percentages of underestimation by the retrieved column density of the actual column densities ($1x10^{17}$ to $1x10^{19}$ $SO_2$ molec. $cm^{-2}$) for windows in the 300-320 nm region and a spectrometer with similar spectral resolution (0.8 nm). Enabling the offset function increased the accuracy of the $SO_2$ dSCDs of many windows compared to the base case (Figs. 2 & 3 (B+O)). The offset function resulted in slightly more windows with accurate dSCDs than the filter for windows with $\lambda_{low}$ <311 nm because the offset function attempts to compensate for all the stray light, not just the stray light originating from >400 nm (Fig. 2 (B+F) & (B+O)). The use of both the offset function and the filter slightly improved the dSCD accuracy for a few windows compared to the filter or offset function alone, mostly for windows with large $\lambda_{high}$ (>324 nm) (Fig. 2 (B+F+O)).

Fitting windows produced more accurate $SO_2$ dSCDs from spectra measured at higher α (90º & 30º) compared to the lowest α (2º & 4º) in the base case (Figs. 2 & 3 (B) & S1). Windows with $\lambda_{low}$ <307 nm underestimated $SO_2$ dSCDs more from the 2º compared to the 30º measurements (Fig. 3 (B)). The spectra collected at higher α are expected to produce more accurate $SO_2$ dSCDs because of the greater UV signal intensity (Fig. 4). The impact of stray light on fits from the lower angle spectra is further increased because the visible light intensity, a potential source of stray light, is the same or higher compared to measurements at higher α (Fig. 4). The difference in the accuracy of $SO_2$ dSCDs between low and high α spectra decreased with the use of the filter or the offset function (Figs. 2-3), an expected result.

Fitting windows with $\lambda_{low}$ >312 nm often overestimated the $SO_2$ dSCDs for all scenarios, as indicated by the green and black areas in Fig. 3, probably because the $SO_2$ absorption features become relatively weak (Fig. 4). Fickel and Delgado Granados (2017) proposed the use of the higher wavelength fitting window of 314.7-326.7 rather than 310-322 nm for $SO_2$ column densities between $10^{17}$ and $10^{18}$ molec. $cm^{-2}$. In contrast, the results of this study found that $SO_2$ dSCDs from the higher range were less accurate than those from the lower range. The threshold for using fitting windows with higher wavelengths due to large optical densities may be greater than $10^{17}$ molec. $cm^{-2}$.

The SO$_2$ dSCDs exhibited a dependence on the features of the SO$_2$ absorption cross-section for $\lambda_{low}$ <307 nm for the base case (Figs. 2-3 (B)) that will be discussed in section 3.3.

## 3.2 Lower Concentration Reference Cell

Figs. 5 and 6 show the SO$_2$ dSCDs and their deviations from the expected value (fit error), respectively, for the lower concentration measurements for all the scenarios. Purple and green areas in Fig. 6 indicate dSCDs were under- and over-estimation, respectively. Black and grey areas indicate dSCDs over- and under-estimated by more than $2.0 \times 10^{16}$ molec cm$^{-2}$, respectively. The SO$_2$ dSCDs from the base case exhibited a dependence on the SO$_2$ absorption that will be discussed in section 3.3. In the base case, the lower concentration measurements had fewer windows that produced accurate SO$_2$ dSCDs compared to the higher concentration measurements (Figs. 2 & 5 (B)). Most of the fitting windows produced SO$_2$ dSCDs that were >100% over- or under-estimated for the lower concentration 2$^o$ spectrum (Figs. 5 & 6 (B) & S1). In contrast, the lower concentration 90$^o$ measurement exhibited accurate SO$_2$ dSCDs for all fitting windows with $\lambda_{low}$ < 311 nm (Fig. S1). This difference highlights that measurements at lower $\alpha$ experience greater inaccuracies from the reduced solar intensity and greater impact of stray light. While the higher concentration dSCDs from the 2$^o$ measurements were consistently underestimated for windows with $\lambda_{low}$ <307 nm, the lower concentration measurements often overestimated the dSCDs (Figs. 5 & 6 (B)). This overestimation in spite of the influence of stray light could be due to interference from O$_3$ since the similarity between the absorption features of SO$_2$ and O$_3$ can introduce instability in the retrieval (Kraus, 2006; Rix et al., 2012). The deviation of the dSCD from the true value can depend on the SO$_2$ concentration rather than exhibiting a consistent bias for a fitting window. The use of the filter or offset function increased the accuracy of the SO$_2$ dSCDs for most windows for spectra measured at angles ≤15$^o$ (Fig. 3 & 6 (B+F), (B+O)). The improved accuracy due to the filter indicates that stray light originating from wavelengths >400 nm significantly decreased the accuracy of the SO$_2$ dSCDs for fitting windows at both lower and higher wavelengths. Unexpectedly, the use of both the filter and offset function for the 30$^o$ measurement reduced the accuracy of the SO$_2$ dSCDs compared to the base case for some windows with $\lambda_{low}$<307 nm and $\lambda_{high}$<320 nm (Fig. 6 (B+F+O)). Since the stray light to signal ratio is expected to be lower for the higher elevation measurements, and the filter already reduced the stray light, the offset function may have incorrectly estimated the relatively small amount of remaining stray light at some wavelengths. The offset function may have added unnecessary freedom to the fit, increasing instability and inaccuracy in the dSCD. Also, the offset function compensates for stray light by assuming the stray light is proportional to the measured intensity (see Eqs. 11-12 in Supplemental). If light from wavelengths outside the fitting window contributes to stray light, this assumption is invalid, and the offset function may increase uncertainty in the fit. The short-pass filter may be the preferred method of reducing the impact of stray light compared to the offset function because the filter directly addresses rather than modelling the source of the problem. However, the problems from using both the filter and offset function can be mitigated by using a fitting window with $\lambda_{low}$>307 nm.

## 3.3 Dependence of the dSCD on the SO$_2$ Absorption Features

In the base case, the SO$_2$ dSCDs exhibited an inverse relationship with the depth of the differential SO$_2$ absorption features for windows with $\lambda_{low}$< 307 nm and $\lambda_{high}$<330 nm for non-zenith measurements (Figs. 2 & 5 (B)). The variation in the SO$_2$ dSCD

as a function of $\lambda_{low}$ from the $\alpha=2^\circ$ measurements, given $\lambda_{high}$ of 315 nm and 324 nm, are shown in Figs. 7 and 8, respectively. The $SO_2$ dSCDs varied up to $3.4\times10^{16}$ and $3.0\times10^{16}$ molec cm$^{-2}$ for a 0.4 nm change in $\lambda_{low}$ for the higher and lower concentration measurements, respectively (Figs. 7-8). Note that for the lower concentration measurement, the difference in the retrieved $SO_2$ dSCDs between using $\lambda_{low}$ of 304 nm and 308.5 nm is an order of magnitude in the base case (Fig. 7). For both

concentrations, using the filter or enabling the offset function reduced the dependence of the dSCDs on $\lambda_{low}$ (Figs. 7-8) and increased the accuracy of many of the low wavelength fitting windows (Figs. 3 & 6). The $SO_2$ dSCD dependency was increased by stray light, exhibiting the greatest underestimation when $\lambda_{low}$ coincided with an $SO_2$ absorption peak. Errors due to stray light are enhanced in wavelength regions where absorption is high (Choudhury and Prayagi, 2015). The measured signal was further reduced surrounding an $SO_2$ absorption peak (e.g., ~304.4 nm) compared to an absorption minimum and stray light

"filled-in" the decreased intensity due to the absorption maxima. If an absorption peak is the strongest $SO_2$ feature included in the fit, the resultant deviation between the modelled and measured spectrum in the peak region requires the fit algorithm to underestimate the $SO_2$ dSCD to minimize the cost function (see Supplemental for fitting algorithm details). The inverse relationship between the dSCD and the $SO_2$ absorption features was strongest at $\lambda_{low}$ <307 nm because absorption was greatest and solar signal was smallest (Figs. 4, 7 & 8). The dSCDs exhibited less dependence on the $\lambda_{low}$ when $\lambda_{low}$ = 307-311 nm due

to increased solar intensity and weaker $SO_2$ absorption (Fig. 4). For both higher and lower concentration measurements, the anti-correlation of the $SO_2$ dSCD in the base case was more pronounced for windows with the $\lambda_{high}$= 324 nm than $\lambda_{high}$= 315 nm (Figs. 7-8).

### 3.4 Fit Uncertainties and Accuracy

The uncertainty in the $SO_2$ dSCD reported by the fitting algorithm and the actual deviation from the expected value shall be

referred to as the "fit uncertainty" and the "fit error," respectively. While the fit uncertainty reported by the retrieval is commonly used as the error on the retrieved dSCD, this uncertainty is not always expected to accurately represent the true error. The divergence is due to factors including assumptions about the independence of errors, and the presence of spectral noise and structures in the fit residual (Stutz and Platt, 1996). Tests of modelled spectra with noise added found that when noise becomes large, the true errors of the retrieved trace-gas coefficient were >10% larger than the retrieved error, and the

difference was proportional to the noise level. Also, the inclusion of random residual structures in the spectra caused the fit uncertainty to underestimate the true error by a factor of 3 (Stutz and Platt, 1996). It is useful to examine which fitting windows exhibited the greatest difference between the fit uncertainty and error. The fit uncertainties from the $2^\circ$ spectrum are shown for the higher and lower concentration measurements in the left column of Figs. 9 and 10, respectively. The fit uncertainties for the base case were the greatest for windows with $\lambda_{low}$ <306 nm and $\lambda_{high}$ < 315 nm, and with $\lambda_{low}$ >312 nm (Figs. 9 & 10 (B)).

The purple and black regions in Figs. 9 and 10 indicate that fit error was greater than the fit uncertainty, and the green regions indicate that fit error was less than fit uncertainty. For the higher concentration measurement in the base case, the fit error was significantly greater than the fit uncertainty (by >2.2×10$^{16}$ molec cm$^{-2}$) for most windows when $\lambda_{low}$ <305 nm (black regions in Fig. 9 (B)), corresponding to fit uncertainties that were 20-50% of the fit errors (Fig. S2). Therefore, fitting windows for the higher concentration measurements in low wavelength regions (impacted by stray light and non-linearity effects) can not only

produce less accurate $SO_2$ dSCDs but also exhibit larger divergences between fit uncertainty and error (Figs. 2, 3 & 9 (B), S2). For the lower concentration measurement, the fit error was $>1.1\times10^{16}$ molec cm$^{-2}$ greater than the fit uncertainty for most windows in the base case (black regions in Fig. 10 (B)). The use of the filter or enabling the offset function reduced the fit uncertainties by up to 50% and decreased the difference between the fit errors and uncertainties, particularly for windows with

$\lambda_{low}$ <309 nm (Figs.9, 10, S2). Note that when the filter or offset function was used, increasing $\lambda_{high}$ > ~ 320 nm or decreasing the $\lambda_{low}$ < ~307 nm decreased the fit uncertainty but not the fit error for some windows (Figs. 6 & 8).

## 4 Summary & Recommendations

Measurements of calibration gas cells with column densities of $2.2\times10^{17}$ and $2.2\times10^{16}$ molec cm$^{-2}$ $SO_2$ were conducted using a mini-MAX-DOAS instrument with an OceanOptics USB2000 spectrometer. In the base case, $SO_2$ dSCDs were least accurate

and had the largest fit uncertainties for fitting windows with $\lambda_{low}$ <307 nm and >312 nm due to stray light and low solar signal, and weak $SO_2$ absorption, respectively. Fitting windows exhibited less accurate $SO_2$ dSCDs for spectra recorded at lower compared to higher α due to reduced UV signal. Therefore, choosing an accurate fitting window is particularly important for measurements at low α. Windows with $\lambda_{low}$ <307 nm generally underestimated $SO_2$ dSCDs from higher concentration measurements for all scenarios. In contrast, many windows with $\lambda_{low}$ <307 nm from the lower concentration measurements

overestimated the $SO_2$ dSCD that were overestimated from the higher concentration measurement. In the base, the $SO_2$ fit uncertainties were significantly less than the actual known fit error for many windows for both concentration measurements. Using the short-pass filter or the offset function increased the accuracy of the $SO_2$ dSCDs, decreased fit uncertainty, and decreased the difference between the fit uncertainty and error compared to the base case for most windows. Some low wavelength windows continued to underestimate the $SO_2$ dSCDs despite the filter for the higher concentration measurements.

The remaining underestimation was probably due to significant stray light originated from <400 nm and non-linearity in the relationship between the column density and measured optical density due to large optical depths of $SO_2$ at these lower wavelengths (e.g., >0.08). A low pass filter with lower cut-off wavelength (i.e., $\lambda_{cut-off}$ = 340 nm) may address the first factor, as may the use of spectrometers with reduced stray light. A spectrometer with an improved spectral resolution should help reduce the impact of the second factor (Kern, 2009). $SO_2$ dSCDs exhibited an inverse dependence on the depth of the

differential features in the $SO_2$ absorption cross-section in the base case. The dependence decreased with the use of the short-pass filter or offset function, implying that stray light contributed to the dependence. Using both the filter and offset function decreased the accuracy of the lower concentration dSCDs of $SO_2$ for some windows with $\lambda_{low}$<307 nm and $\lambda_{high}$<320 nm compared to the base case. Increasing the $\lambda_{high}$ greater than ~ 320 nm tended to decrease the fit uncertainty but not necessarily the fit error for measurements with the filter or offset function.

Note that this study focused on the impact of two retrieval parameters (the fitting window wavelength and offset function) but that several other parameters can be varied in the $SO_2$ dSCD fit. These parameters include the order of the DOAS and offset function polynomials and the choice of the literature cross-sections for the trace gases. The DOAS analysis could also be expanded to include a correction to reduce impacts of the wavelength dependence of the Ring effect in the near UV due to aerosol and multiple Rayleigh scattering, as described in Langford et al. (2007). Additional factors that could impact the

retrieved dSCD include the solar zenith and azimuth angles during measurement. A limitation of this study is the lack of measurements at high solar zenith angles (near dawn and dusk) when the SCDs of $O_3$ are larger and change rapidly with time. In such cases, fit accuracy may benefit from extending the upper limit of the fit window to allow better discrimination between the differential absorption features in the $O_3$ and $SO_2$ cross-sections. Future studies could repeat these experiments by

measuring at different solar geometries and varying the other fit parameters. Also, $SO_2$ the column densities measured in this study were chosen to be representative of a range typical of polluted urban settings. Greater $SO_2$ column densities ($>1\times10^{18}$ molec. cm$^{-2}$) can be observed in volcanic areas and close to major industrial sources; discussions of retrieving such greater $SO_2$ column densities can be found in Bobrowski et al. (2010) and Fickel and Delgado Granados (2017).

Based on the results of this study, it is recommended that fitting windows for $SO_2$ have $\lambda_{low}$ >307 nm to avoid the effects of

stray light, low solar signal, and, for higher column densities, effects of non-linearity between the measured optical depth and the column density for optical densities >>0.05, and $\lambda_{low}$ <312 nm because of weak $SO_2$ features. Fitting windows are recommended to have $\lambda_{high}$ less than ~320 nm to avoid increased underestimation of the fit error by the fit uncertainty unless there are concerns about interference by large $O_3$ absorptions, such as at high SZA. A fitting window should not be chosen only because it has a smaller fit uncertainty since it does not guarantee a more accurate dSCD. A short-pass filter with a cut-

off close to the $\lambda_{high}$ of the $SO_2$ fitting window improves the accuracy of MAX-DOAS $SO_2$ measurements. In the absence of a filter or if a filter would conflict with other species to be determined (e.g., $NO_2$), the offset function should be used to compensate for stray light. Even in the case that $SO_2$ and $NO_2$ are to be fit simultaneously, a filter with $\lambda_{cut-off}$ = 550 nm may reduce stray light. A short-pass filter may be preferred over the offset function for reducing stray light impacts because the filter removes stray light while the offset function mathematically compensates for stray light by assuming it is proportional

to the measured intensity (see Eqs. 11-12 in Supplemental). The offset function may increase fit error if this assumption is invalid or if little stray light is present. If a short-pass filter or the offset function is used, the 307.5-319 nm fitting window for mini-MAX-DOAS measurements of $SO_2$ is recommended. Ultimately, the use of higher quality spectrometers with reduced stray light and improved spectral resolution for MAX-DOAS measurements is desirable, but a greater expense compared to the low-cost spectrometer used in this study.

*Author Contributions.* ZYWD: MAX-DOAS study concept, design, investigation and data analysis, data visualization, and writing of manuscript and modifications of the same with contribution from co-author RM. RM: MAX-DOAS supervision.

*Competing Interests.* The authors declare that they have no conflict of interest.

Data availability. The MAX-DOAS data collected from this study are publicly available with the following DOI: https://doi.org/10.5683/SP2/4XQZS8.

*Acknowledgements.* This study was supported by the Natural Sciences and Engineering Research Council of Canada (NSERC) (Discovery grant no. RGPIN-2018-05898) and Collaborative Research and Training Experience Program (CREATE) Integrating Atmospheric Chemistry and Physics from Earth to Space (IACPES) (grant no. 398061-2011).

10    **Appendix A** List of symbols and acronyms used in this paper.

| Acronym | Expansion |
| --- | --- |
| $\alpha$ | Viewing Elevation Angle |
| $\lambda_{cut\text{-}off}$ | Cutoff wavelength of Short-pass Filter |
| $\lambda_{high}$ | Upper Limit Wavelength of Fitting Window |
| $\lambda_{low}$ | Lower Limit Wavelength of Fitting Window |
| (B) | Base Case Measurement (No Filter and Offset Function Disabled) |
| (B+F) | Measurement with Short-Pass Filter |
| (B+F+O) | Measurement with Short-Pass Filter Fit using Offset Function |
| (B+O) | Measurement with Fit using Offset Function |
| dSCD | Differential Slant Column Density |
| FRS | Fraunhofer Reference Spectrum |
| HCHO | Formaldehyde |
| MAX-DOAS | Multi-Axis Differential Optical Absorption Spectroscopy |
| molec cm$^{-2}$ | Molecules per square centimetre |
| nm | nanometres |
| $NO_2$ | Nitrogen Dioxide |
| $O_3$ | Ozone |
| ppb | Parts Per Billion |
| SCD | Slant Column Density |
| $SO_2$ | Sulphur Dioxide |
| UV | Ultraviolet |
| VCD | Vertical Column Density |

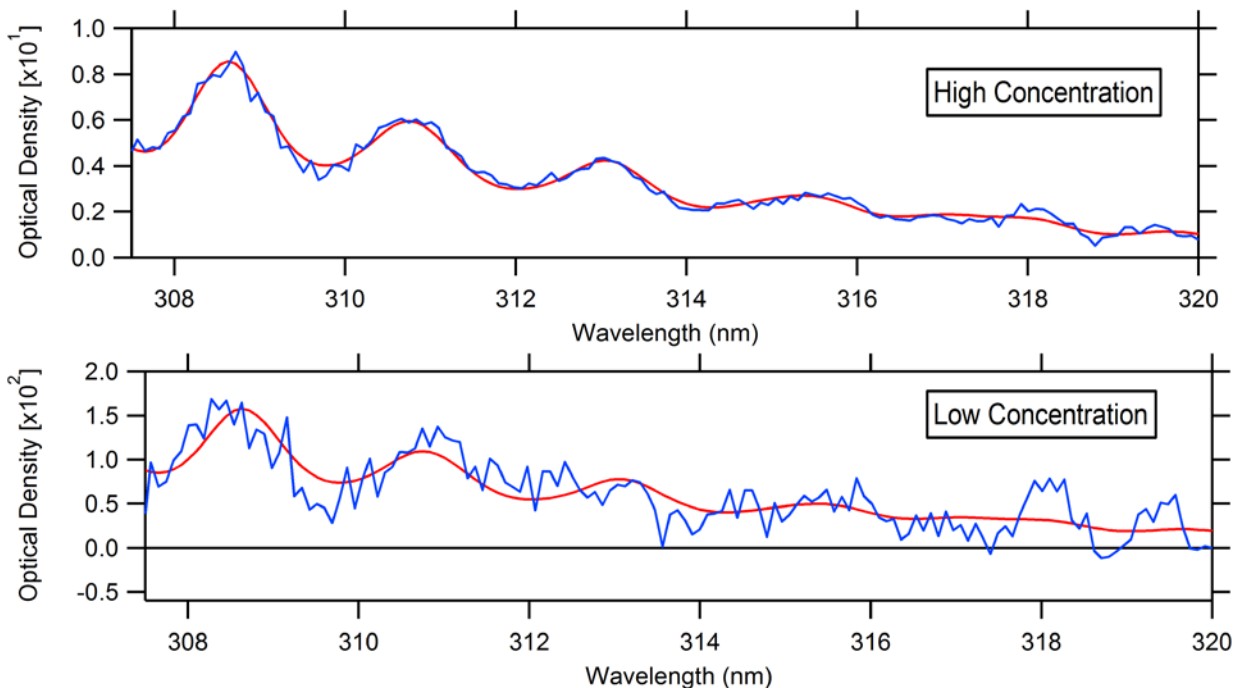

**Figure 1** Examples of spectral retrievals of SO$_2$ from the base case (no filter and offset function disabled) from spectra measured at 2° viewing elevation angle using the fitting window 307.5-320 nm. Retrieved dSCDs were 2.23(+/-0.08)×10$^{17}$ molec cm$^{-2}$ and 4.10(+/-0.66)×10$^{16}$ molec cm$^{-2}$ for the higher and lower concentration measurements, respectively.

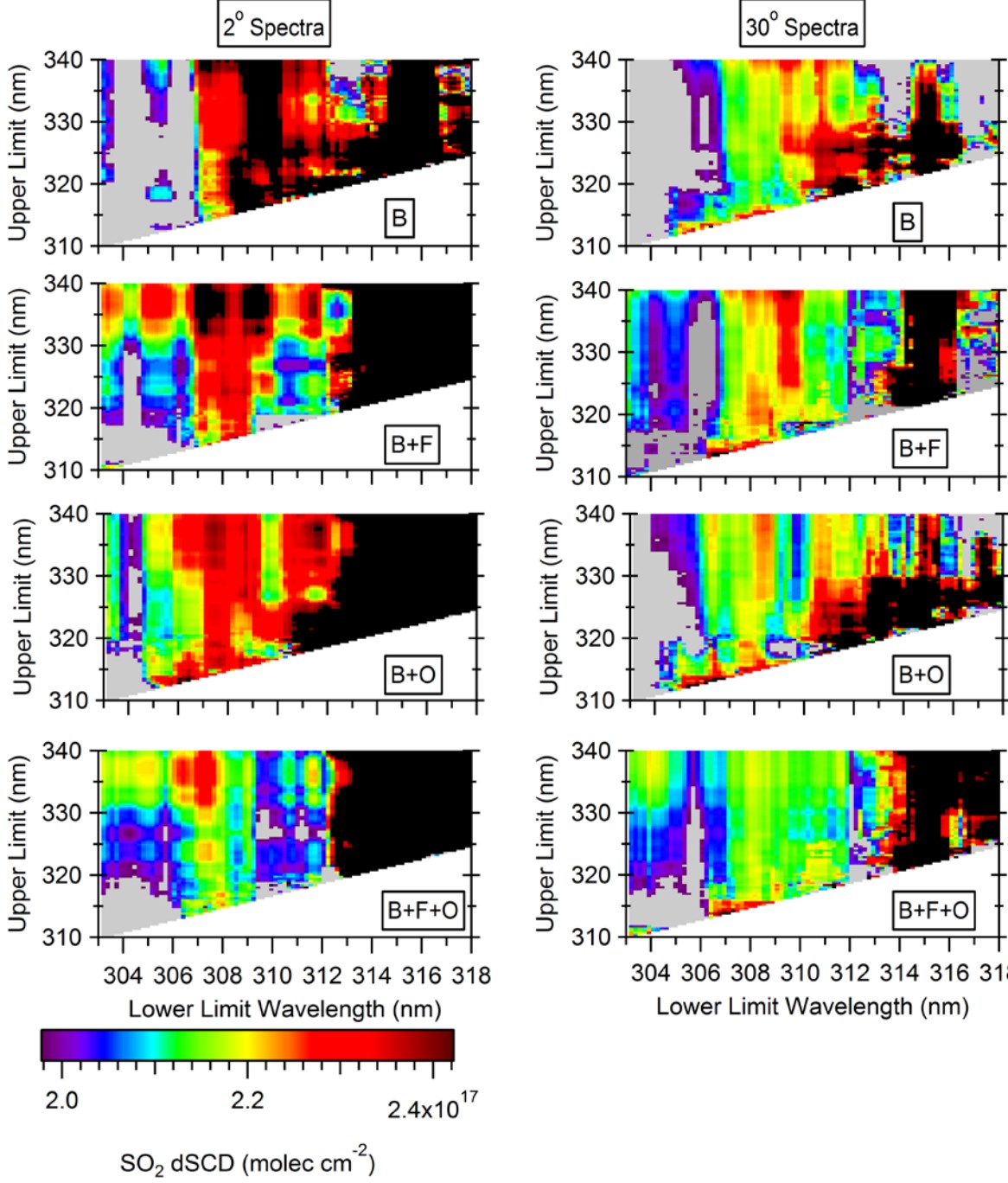

**Figure 2** SO$_2$ dSCDs fit from higher concentration measurements at 2° (left) and 30° (right) elevation angles for the base case (B), with offset (B+O), with filter (B+F), and with filter and offset (B+F+O). Grey and black areas indicate dSCDs were >10% less and >10% more than the expected value, respectively. The true value of the cell is 2.2×10$^{17}$ molec cm$^{-2}$ (yellow).

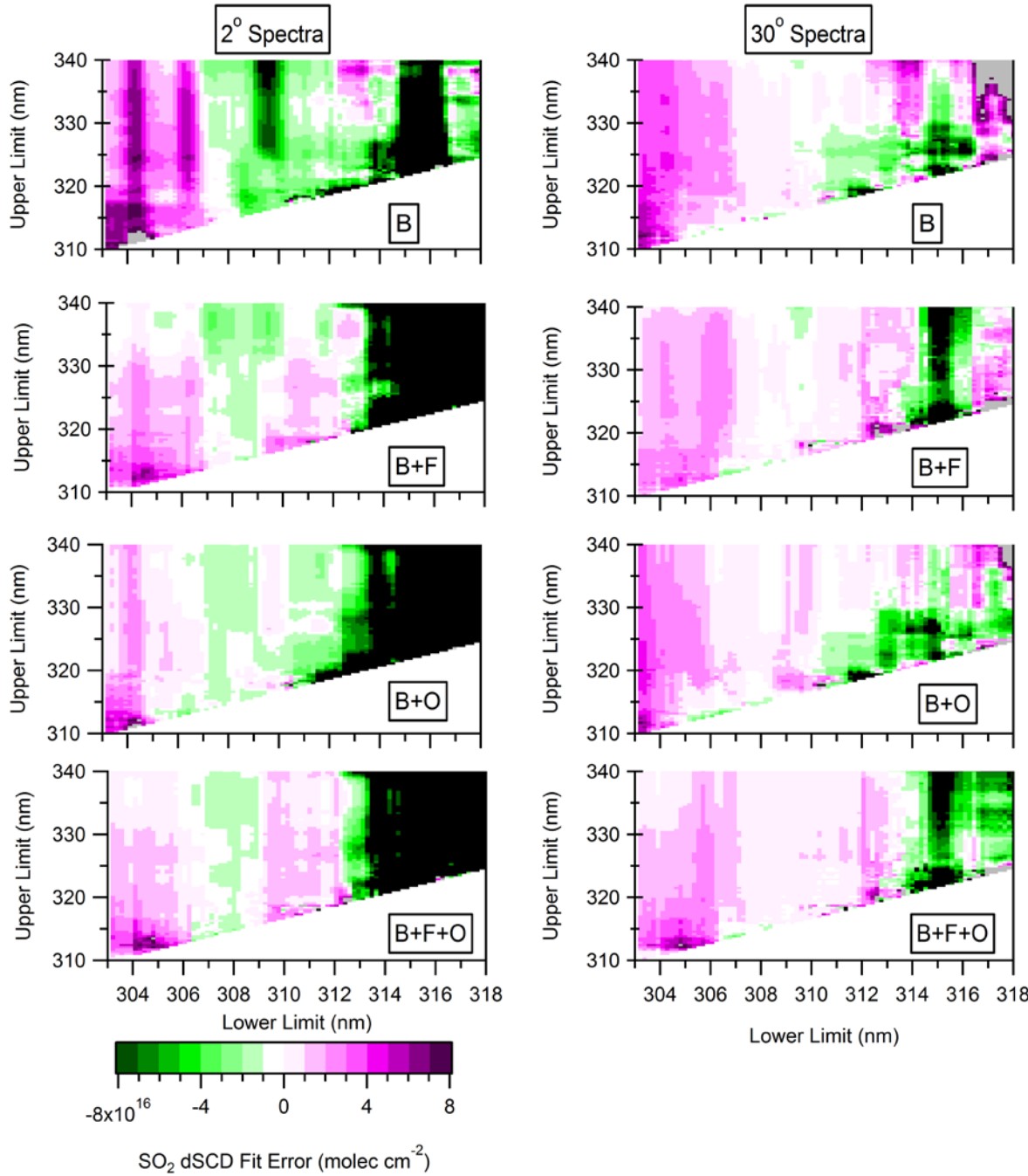

**Figure 3** Higher concentration fit errors (deviations of $SO_2$ dSCDs from the expected value of $2.2\times10^{17}$ molec cm$^{-2}$) from the measurements at 2° (left) and 30° (right) elevation angles for the base case (B), with offset (B+O), with filter (B+F), and with filter and offset (B+F+O). Purple and green areas indicate under- and over-estimation of the expected value, respectively. Black and grey areas indicate dSCDs over- and under-estimated by more than $8.0\times10^{16}$ molec cm$^{-2}$, respectively.

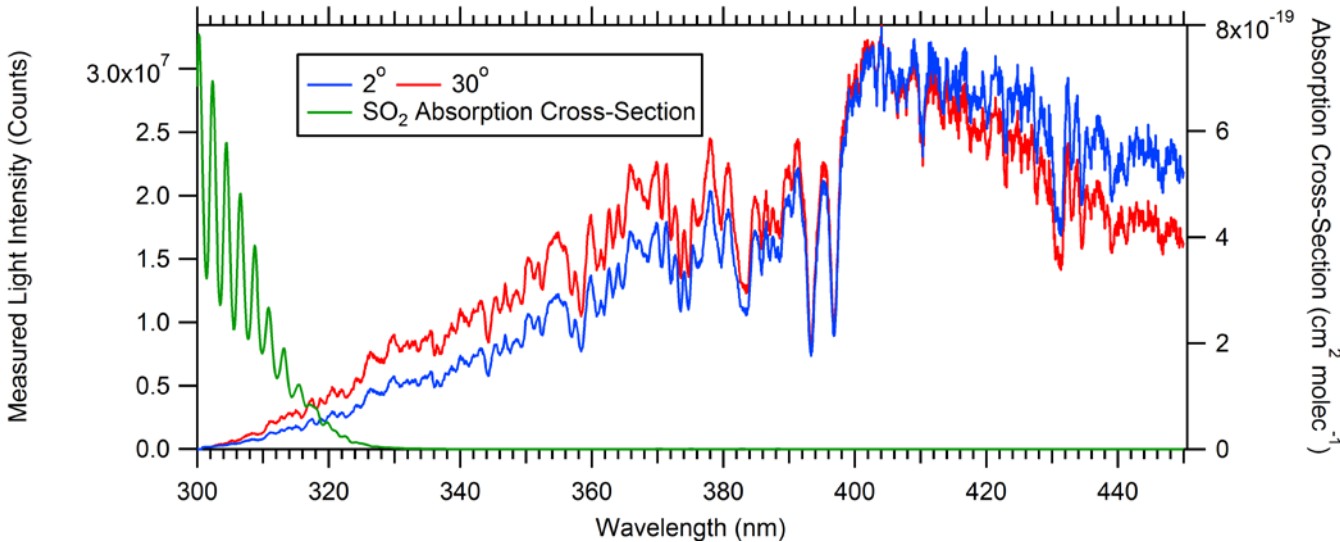

**Figure 4** Comparison of the measured spectral intensity for the 2º and 30º viewing elevation angle spectra with the lower concentration cell without the short-pass filter, and the absorption cross-section of $SO_2$ smoothed to the spectral resolution of the instrument.

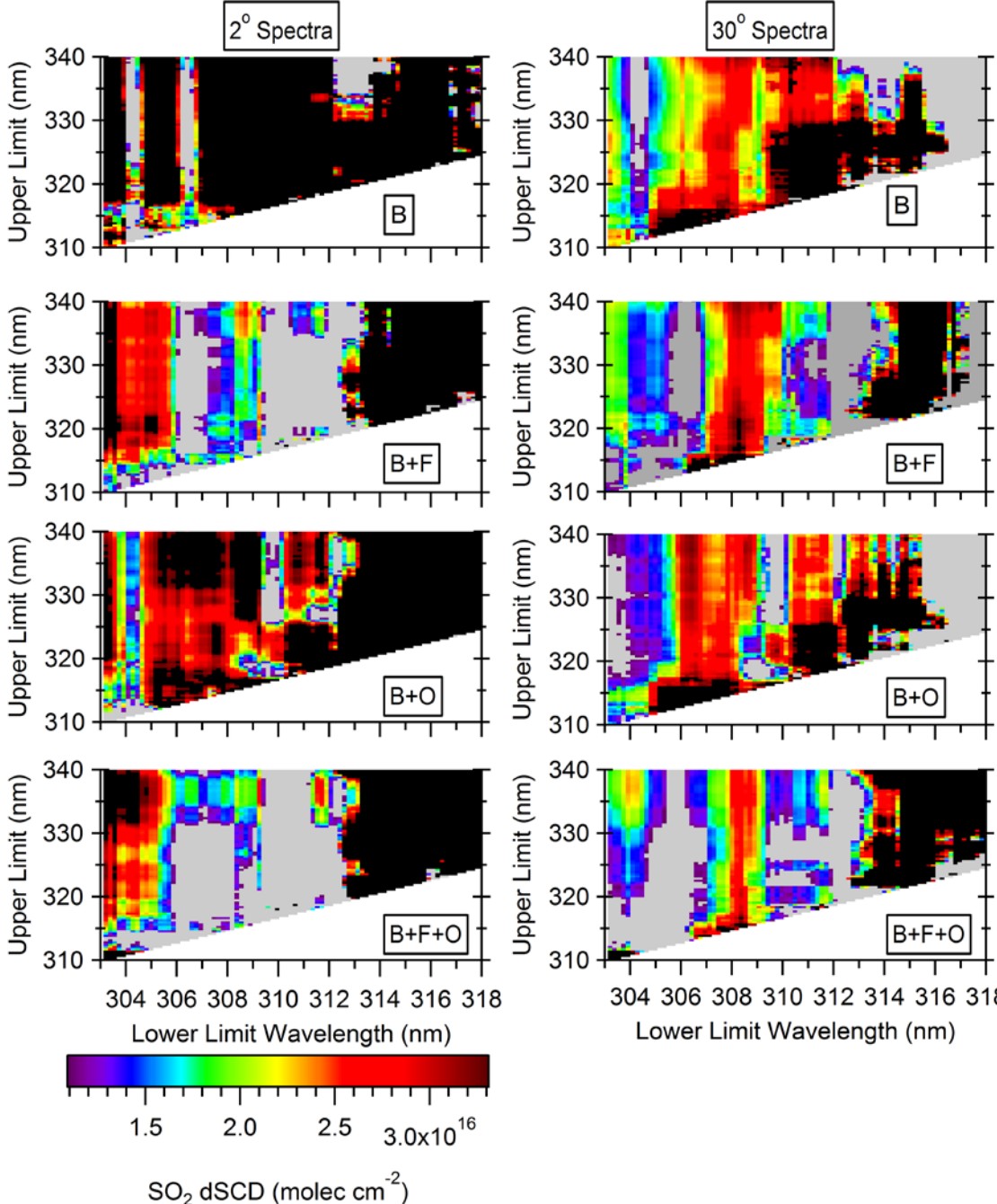

**Figure 5** SO$_2$ dSCDs fit from the lower concentration measurements at 2° (left) and 30° (right) elevation angles for the base case (B), with offset (B+O), with filter (B+F), and with filter and offset (B+F+O). Grey and black areas indicate dSCDs that were <50% less and >50% more than the expected value, respectively. The true value of the higher concentration cell is 5  2.2×10$^{16}$ molec cm$^{-2}$ (yellow).

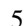

25

30

**Figure 6** Lower concentration fit errors (deviations of $SO_2$ dSCDs from the expected value of $2.2{\times}10^{16}$ molec cm$^{-2}$) from the measurements at 2$^\circ$ (left) and 30$^\circ$ (right) elevation angles for the base case (B), with offset (B+O), with filter (B+F), and with filter and offset (B+F+O). Purple and green areas indicate dSCDs were under- and over-estimation, respectively. Black and grey areas indicate dSCDs over- and under-estimated by more than $2.0{\times}10^{16}$ molec cm$^{-2}$, respectively.

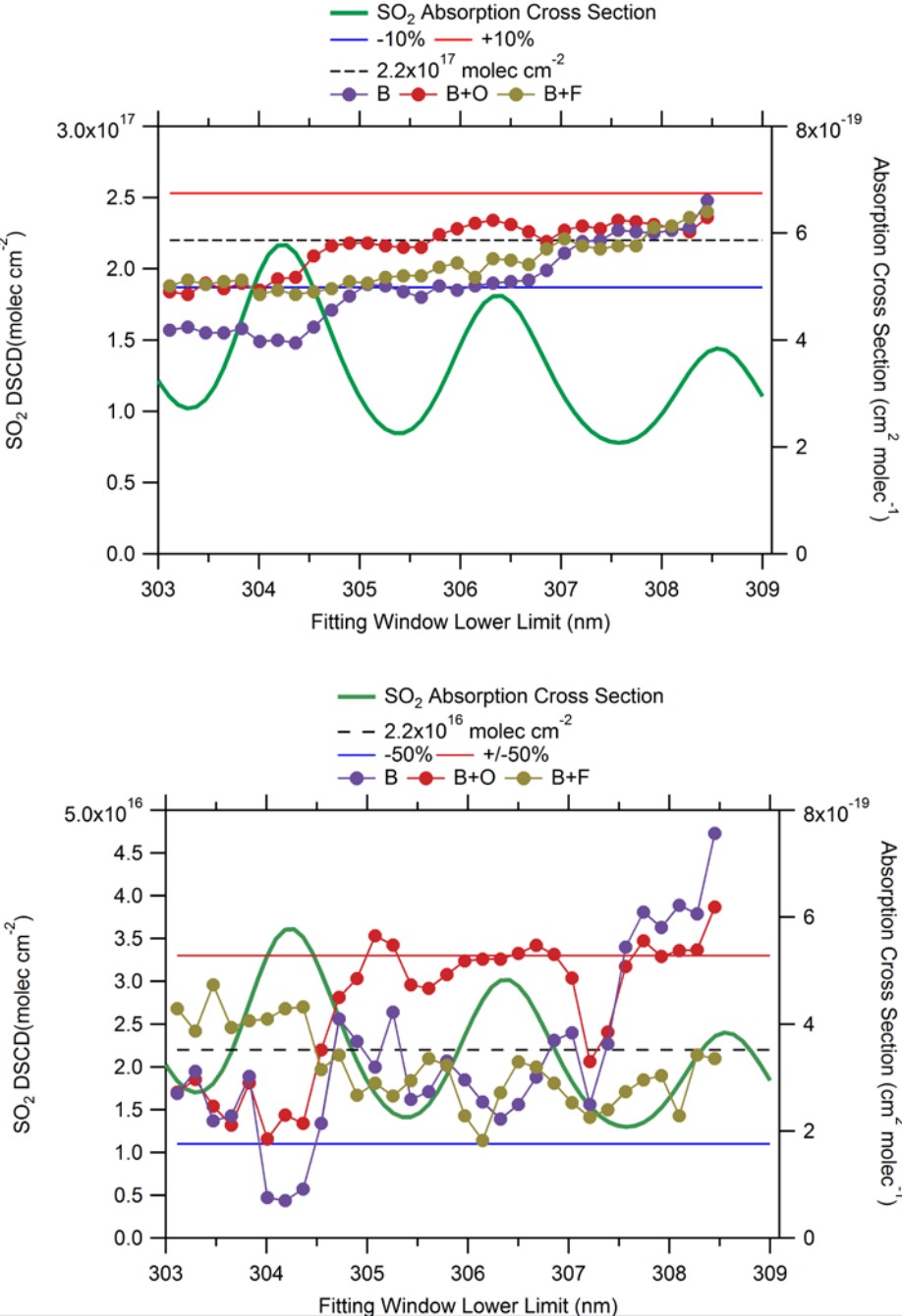

**Figure 7** SO$_2$ absorption cross-section and variation in the SO$_2$ dSCD with $\lambda_{low}$ with $\lambda_{high}$ = 315 nm for higher (top) and lower (bottom) concentration measurements for the base case (B), with offset (B+O), and with filter (B+F).

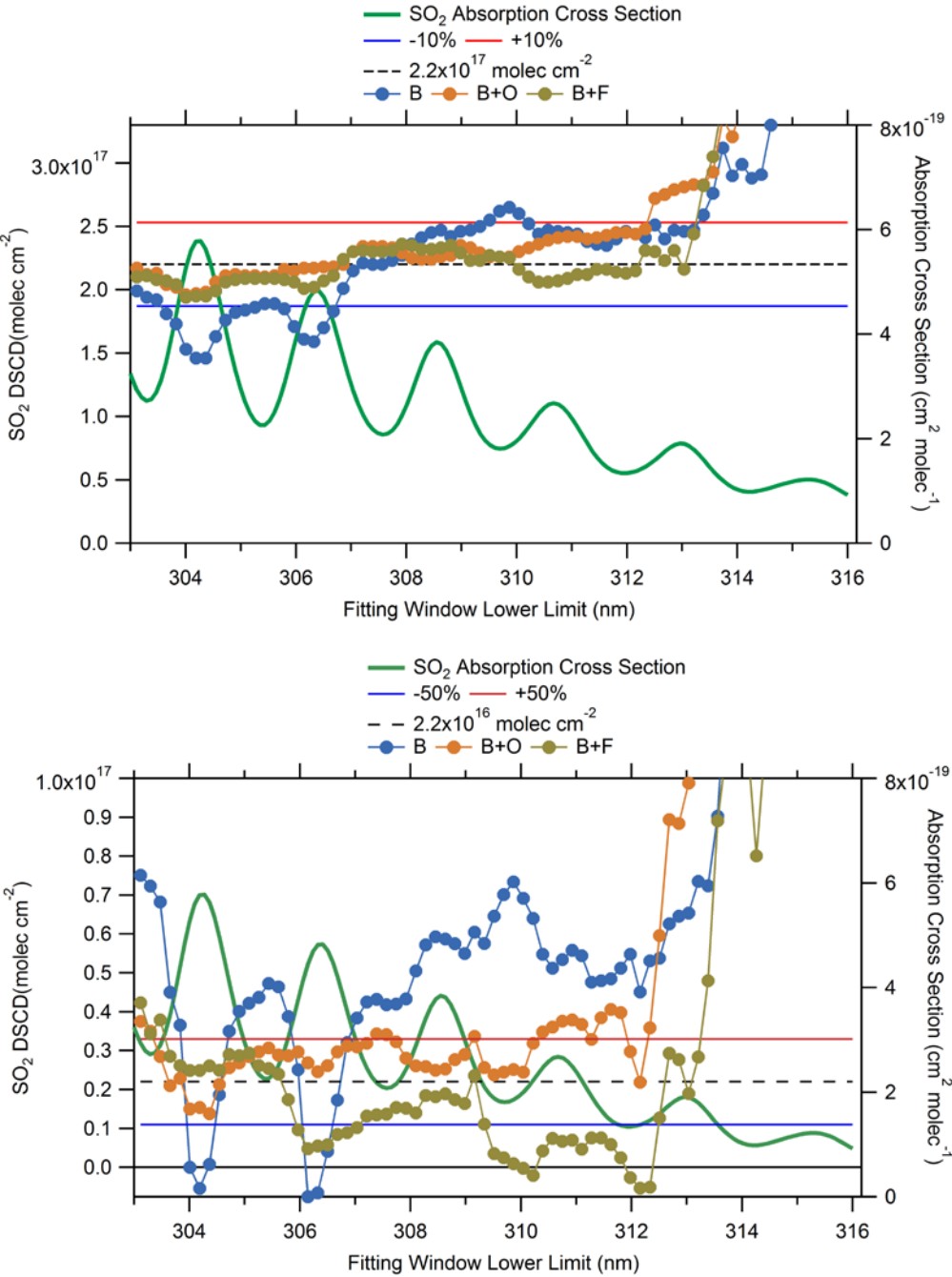

**Figure 8** SO$_2$ absorption cross-section and variation in the SO$_2$ dSCD with $\lambda_{low}$ with $\lambda_{high}$= 324 nm for higher (top) and lower (bottom) concentration measurements for the base case (B), with offset (B+O), and with filter (B+F).

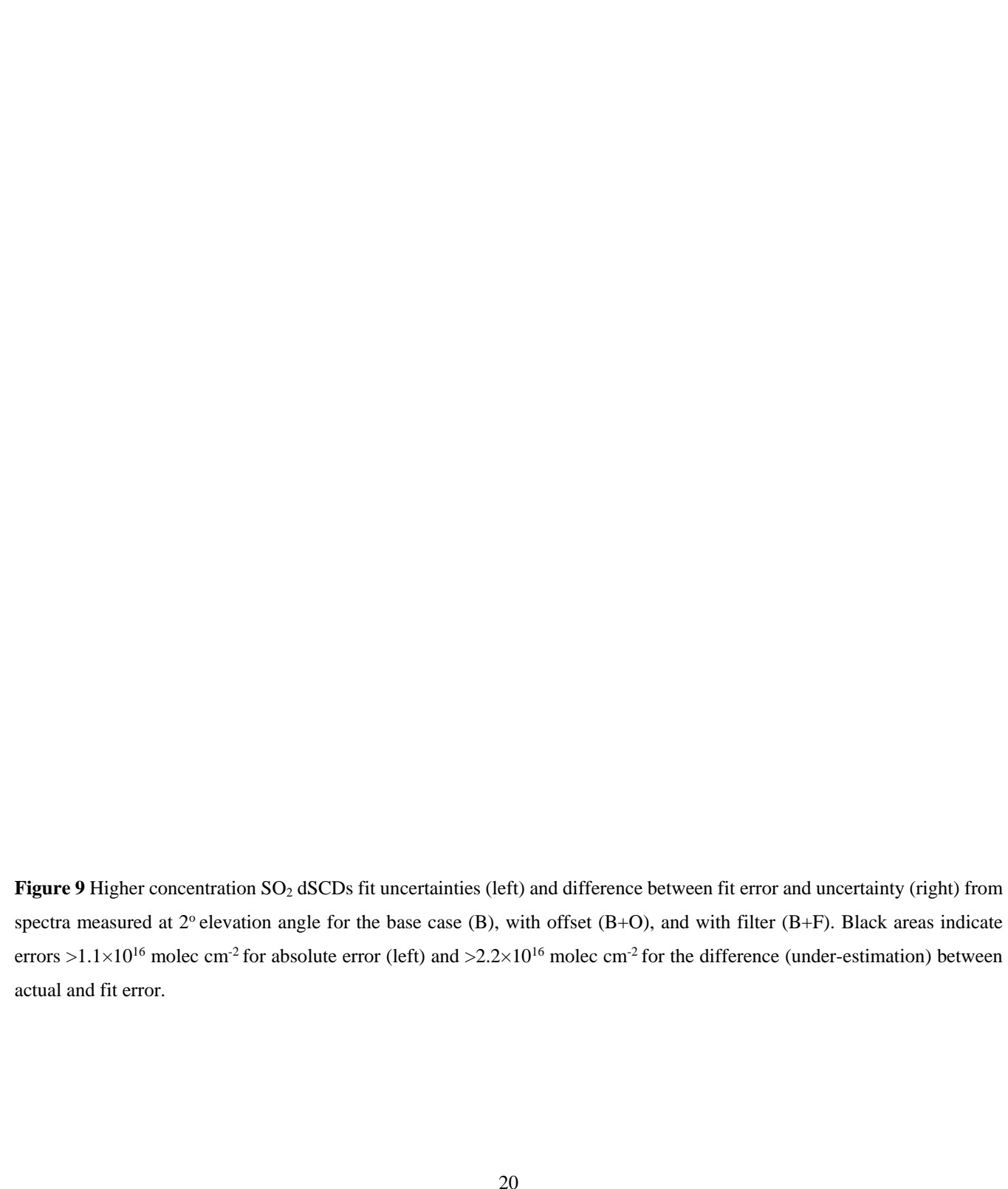

**Figure 9** Higher concentration $SO_2$ dSCDs fit uncertainties (left) and difference between fit error and uncertainty (right) from spectra measured at 2° elevation angle for the base case (B), with offset (B+O), and with filter (B+F). Black areas indicate errors $>1.1\times10^{16}$ molec cm$^{-2}$ for absolute error (left) and $>2.2\times10^{16}$ molec cm$^{-2}$ for the difference (under-estimation) between actual and fit error.

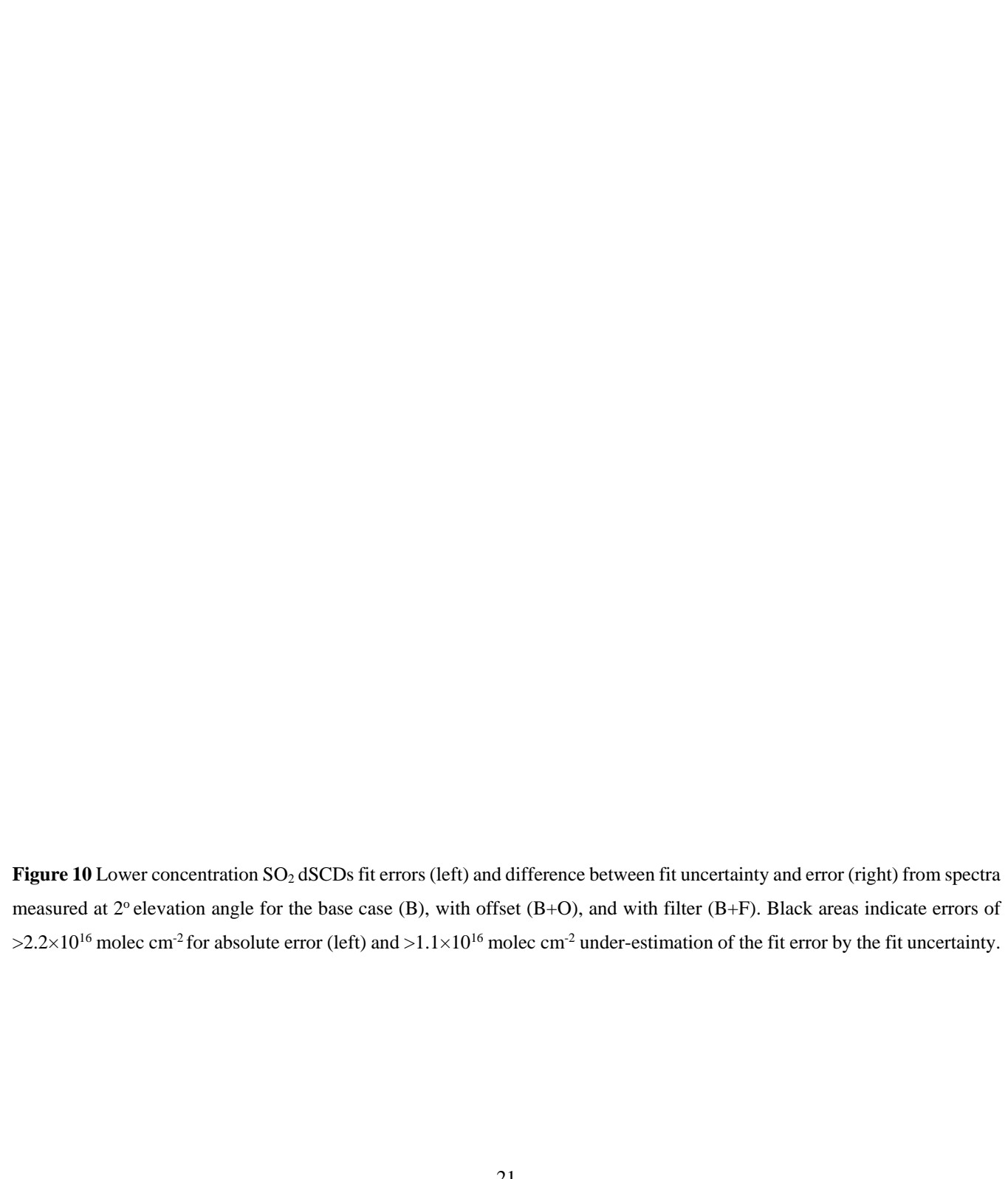

**Figure 10** Lower concentration SO$_2$ dSCDs fit errors (left) and difference between fit uncertainty and error (right) from spectra measured at 2° elevation angle for the base case (B), with offset (B+O), and with filter (B+F). Black areas indicate errors of >2.2×10$^{16}$ molec cm$^{-2}$ for absolute error (left) and >1.1×10$^{16}$ molec cm$^{-2}$ under-estimation of the fit error by the fit uncertainty.

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
