# Peer review of "Recommendations for Spectral Fitting of SO2 from mini-MAX-DOAS Measurements"

_Atmospheric Measurement Techniques, 2019_

## Referee Comment (RC1) · Anonymous Referee #1 · 27 Feb 2020

Title: Recommendations for spectral fitting of SO2 from MAX-DOAS measurements.

The aims of the article is to find the best fitting window for SO2. Sulfur bioxide is a gas that is difficult to measure due to the proximity of its absorption band at low wavelengths, the main problems encountered: low ultraviolet radiation and stray light. The article is well structured and clear in its dissertation, considering the complexity of the research topic. The researchers working with SO2 will find this article and its recommendations on the fitting window, errors and uncertaries, very useful.

 c General consideration

1) How would you use the recommendations discussed in the article with satellite com-

parisons? 2) Have you tried the 315-325 nm window fit?

3) Have you thought in a comparison, using the window fit recommend, with satellite? OMI? TROPOMI?

4) Is it necessary to correct the ring effect if active DOAS is used?

5) It would be very useful to include in the article measurement of calibration cells with high SO2 concentrations (∼1e19), applying the recommendations presented in the article and discussing the results.

• Specific consideration

Introduction Line 34: Rix et al., 2012, spectral range fit between 315 and 326 nm.

Line 43: verify reference "Wang and Christopher, 2003", is it correct?

Methods Line 55: Instrument resolution (FWHM) are calculated or is the information provided by the company (in this case, Ocean Optics)? The company provides an average resolution of the entire spectral range of the insturment and not in the specific absorption range of a given compound (e.g. SO2, NO2, etc..).

Line 58: "using the DOASIS software package". Why using DOASIS? In my opinion, QDOAS is currently the best and most widely used software for the fitting of contaminating compounds in the atmosphere with ground-based techniques.

Line 61-62: Be careful to use high and low concentration referring to the calibration cells. I disagree in considering a concentration of ∼1e17 as "high". This is relative. What are you talking about? Contaminated areas? industrial areas? volcanic areas? Megacities? The SO2 concentrations of ∼ 1e17 molec/cm2 (occasionally ∼1e18) are "high" considering anthropogenic contaminated areas, such as megacities or/and industrial areas. Volcanic SO2 concentrations can reach ∼1e19 molec/cm2. Using instruments such as UV camera, very concentrated calibration cells are needed (∼1e19 molec/cm2), to be able to calibrate the instrument for SO2 concentrated plumes. I

suggest specifying the "high concentrations" in the atmosphere of polluted areas, like megacities.

Linea 63: "These SCDs would be equivalent to an air mass with SO2 mixing ratios of 87 and 8.7 ppb. . .". Is the calculation correct? How did you calculate these values?

• Final consideration

Personally, I tried to apply the recommendations, discussed in the article, on the best SO2 fitting windows. I compared, using MAX-DOAS and Direct Sun techniques, the 307.5-319 nm window fit with other windows fit used in the literature. The 307.5-319 nm window fit work well in anthropogenic contaminated areas, I was able to recover more spectra and obtaining a lower RMS than other windows fit. On the other hand, the results obtained by testing the recommended window fit with very concentrated calibration cells (∼1e19), greatly underestimate the real concentration inside the cells. I consider the article important for the scientific community and I recommend publication in AMT. I therefore recommend minor revisions.

Please also note the supplement to this comment:
https://www.atmos-meas-tech-discuss.net/amt-2019-420/amt-2019-420-RC1-supplement.pdf

**Supplement:**

**Title:** Recommendations for spectral fitting of $SO_2$ from MAX-DOAS measurements.

The aims of the article is to find the best fitting window for $SO_2$. Sulfur bioxide is a gas that is difficult to measure due to the proximity of its absorption band at low wavelengths, the main problems encountered: low ultraviolet radiation and stray light.
The article is well structured and clear in its dissertation, considering the complexity of the research topic. The researchers working with $SO_2$ will find this article and its recommendations on the fitting window, errors and uncertaines, very useful.

- **General consideration**

1) How would you use the recommendations discussed in the article with satellite comparisons?
2) Have you tried the 315-325 nm window fit?

3) Have you thought in a comparison, using the window fit recommend, with satellite? OMI? TROPOMI?

4) Is it necessary to correct the ring effect if active DOAS is used?

5) It would be very useful to include in the article measurement of calibration cells with high $SO_2$ concentrations ($\sim 10^{19}$), applying the recommendations presented in the article and discussing the results.

- **Specific consideration**

**Introduction**
Line 34: Rix et al., 2012, spectral range fit between 315 and 326 nm.

Line 43: verify reference "Wang and Christopher, 2003", is it correct?

**Methods**
Line 55: Instrument resolution (FWHM) are calculated or is the information provided by the company (in this case, Ocean Optics)? The company provides an average resolution of the entire spectral range of the insturment and not in the specific absorption range of a given compound (e.g. $SO_2$, $NO_2$, etc..).

Line 58: "using the DOASIS software package". Why using DOASIS? In my opinion, QDOAS is currently the best and most widely used software for the fitting of contaminating compounds in the atmosphere with ground-based techniques.

Line 61-62: Be careful to use high and low concentration referring to the calibration cells. I disagree in considering a concentration of ~$10^{17}$ as "high". This is relative. What are you talking about? Contaminated areas? industrial areas? volcanic areas? Megacities?
The $SO_2$ concentrations of ~ $10^{17}$ molec/cm$^2$ (occasionally ~$10^{18}$) are "high" considering anthropogenic contaminated areas, such as megacities or/and industrial areas. Volcanic $SO_2$ concentrations can reach ~$10^{19}$ molec/cm$^2$. Using instruments such as UV camera, very concentrated calibration cells are needed (~$10^{19}$ molec/cm$^2$), to be able to calibrate the instrument for $SO_2$ concentrated plumes. I suggest specifying the "high concentrations" in the atmosphere of polluted areas, like megacities.

Linea 63: "These SCDs would be equivalent to an air mass with $SO_2$ mixing ratios of 87 and 8.7 ppb…". Is the calculation correct? How did you calculate these values?

- **Final consideration**

Personally, I tried to apply the recommendations, discussed in the article, on the best $SO_2$ fitting windows. I compared, using MAX-DOAS and Direct Sun techniques, the 307.5-319 nm window fit with other windows fit used in the literature. The 307.5-319 nm window fit work well in anthropogenic contaminated areas, I was able to recover more spectra and obtaining a lower RMS than other windows fit.
On the other hand, the results obtained by testing the recommended window fit with very concentrated calibration cells (~$10^{19}$), greatly underestimate the real concentration inside the cells.
I consider the article important for the scientific community and I recommend publication in AMT. I therefore recommend minor revisions.

---

## Referee Comment (RC2) · Christoph Kern (Referee) · 4 Mar 2020

**Christoph Kern (Referee)**

ckern@usgs.gov

Received and published: 4 March 2020

Review

Recommendations for spectral fitting of SO2 from MAX-DOAS measurements

Zoë Davis and Robert McLaren

**General Remarks**

This article describes the results of some basic experiments aimed at determining an ideal wavelength window for the analysis of MAX-DOAS measurements of sulfur dioxide (SO2). The authors placed gas cells containing known amounts of SO2 in

front of a MAX-DOAS instrument while recording spectra of an unpolluted atmosphere. When analyzed relative to spectra obtained without using the cells, these measurements should yield the known SO2 column densities of the gas cells. Any deviation from these known values is attributable to errors induced during the fitting procedure.

In an effort to find an optimal setup for the DOAS fit, the authors varied the lower and upper boundaries of the fit wavelength window in an approach called 'retrieval interval mapping' (Vogel et al., 2013). They concluded that using the range 307.5 to 319 nm generally yielded the most accurate results in the considered cases. In addition, the effects of suppressing stray light either by use of a short-pass coloredglass filter in the spectrometer's entrance optics or by accounting for it through inclusion of an offset polynomial in the DOAS fit was assessed. Both strategies appeared to generally improve the accuracy of the results.

This article is well-written and informative, and the results are useful for the atmospheric sciences and volcanic gas communities. My only concern is that the considered experiments may be too limited in scope to be able to support the relatively broad conclusions regarding the ideal fit window. In particular, the authors do not consider the impact of a rapidly varying ozone slant column density (SCD) as it occurs early or late in the day, instead only analyzing spectra recorded around solar noon. Also, the authors apply a relatively simple Ring correction in their retrievals, when a more sophisticated approach may improve the fit quality and/or allow the width of the fit window to be increased, thus improving the robustness of the fit results. Finally, the authors only tested a single spectrometer and it's not entirely clear how these results apply to other, possibly more sophisticated MAX-DOAS instruments. Below, I have listed some recommendations on how the authors might improve their manuscript with regards to these points. Once these comments have been considered, I recommend the article be published in Atmospheric Measurement Techniques.

Specific Issues
It is my experience that imperfect representation of the differential ozone absorption in the 300 to 340 nm region in the DOAS model can lead to errors in retrieved SO2 column densities. These effects are not captured by the authors' experiments because they did not assess the effect of a varying solar zenith angle (SZA) and hence a change in the ozone SCD on their results. My worry is that these might affect the recommendations for wavelength fit window, possibly leading the authors to consider broadening the window to allow for better discrimination between SO2 and O3. To address this issue, the experiments could be repeated for a time early in the morning or late in the evening when the O3 SCD is changing rapidly with time. This then leads to a mismatch in O3 SCD between 2-degree and 90-degree MAX-DOAS observations, a mismatch that needs to be accounted for by the O3 cross-section included in the DOAS fit. As the currently discussed experiments were all conducted around solar noon, the impact of O3 in the spectra is likely negligible, but the results are not necessarily valid for measurements made throughout the day.

A more careful consideration of the Ring effect would also be worthwhile. The SO2 and O3 differential absorption features are of a similar bandwidth as the Fraunhofer lines in the solar spectrum. Hence, an imperfect removal of the filling-in of these lines by inelastic Raman-scattered radiation could potentially interfere with the SO2 retrieval. I recommend the authors review the literature with regards to state-of-the-art Ring correction, in particular focusing on an additional dependency of the Ring effect on wavelength (Vountas et al., 1998 Langford et al., 2007), the potential impact of vibrational Raman scattering (Lampel et al., 2015), and the effect of the broad-band shape of the solar spectrum as it reaches the Earth's surface (Lampel et al., 2017). If each of these effects are properly accounted for, it may be possible to extend the width of the fit window beyond the authors may find that these effects are of second order importance and their consideration does not improve the SO2 retrieval. But either way, I believe they should be tested in order to ensure that the authors recommendations reflect the state-of-the-art.

**AMTD**
Finally, it's a bit unclear to me how representative the results are for MAX-DOAS instruments in general. The authors use an Ocean Optics USB2000 spectrometer for their measurements. This instrument is very common and therefore represents a good choice for such a study. However, it is my experience that these relatively economical instruments suffer from a relatively poor stray-light rejection. Therefore, it is no surprise that suppression of stray light using an optical filter and/or accounting for it in the DOAS fit improves the results of the retrieval. At the same time, these results and recommendations may not apply to other, higher quality instruments. It would be of great value if the authors were able to compare their results with those obtained using a higher-grade spectrometer. If this is not possible, the authors should consider explicitly narrowing the scope of their manuscript to reflect the fact that the experiments were all made using this one type of instrument. For example, the authors might include the terms 'low-cost' or 'miniature' in the title and mention the make and model of the spectrometer in the abstract and prominently throughout the manuscript.

Minor issues and corrections:

P1L11 – I recommend rewording "the dSCDs also exhibited an inverse relationship with the DEPTH OF THE DIFFERENTIAL FEATURES IN THE SO2 absorption cross-section..."

P1L15 – "... dependence on the SO2 absorption features SUGGESTING THAT THE RADIANCE AT SHORTER WAVELENGTHS WAS increased by stray light..."

P1L18 and P7L25 – The uncertainty reported by the fit is not necessarily expected to be an accurate measure of the errors of the results. This is discussed by Stutz and Platt (1996). Please incorporate this information into your discussion.

P2L6 - I suggest adding "typically" before "uses the SO2 B band..."

P2L15 – In my experience, it can be beneficial to include wavelengths with weak or negligible SO2 absorption in the fit if the extended wavelength range allows for better
discrimination between SO2 and other aspects of the DOAS model (Ozone absorption, Ring effect).

P3L6 - "... may NOT BE IDEAL for smaller..."

P3L23 – please remove the superscript formatting on the "C".

P3L30 – Please clarify how the 87 ppb were arrived at. I think this is assuming that the 30 degree spectra were evaluated relative to a reference recorded looking towards the zenith through the same atmospheric boundary layer, correct?

P3L31 – Please include the time at which the measurements were made. This will allow the readers to deduce the solar zenith angle. How far apart in time where the 2-degree and 90-degree measurements made?

P4L2 – In the future, you might consider using a Hoya U330 filter. I believe this has better rejection of NIR radiation which can cause stray light in the spectrometer.

P5L23 – Wavelengths longer than 324 nm are commonly used in volcanic gas measurements. While they admittedly often encounter larger SO2 SCDs, it's not clear to me that these wavelengths can be discounted across the board for MAX-DOAS measurements simply based on the increased DOF. See comment above on improved discrimination between SO2, ozone and Ring features.

P5L29 – It's not clear to me how increased Rayleigh scattering due to higher air pressure would preferentially remove shorter UV wavelengths. Is there a citation that you could provide for this? Couldn't one just as easily argue that increased Rayleigh scattering of previously unscattered sunlight would increase the measured radiance at shorter wavelengths? I suspect that another effect or combination of effects is responsible, possibly having to do with radiation being removed from the atmospheric half-sphere by absorption on the ground. In this case, it might be the wavelengthdependence of the ground albedo that is responsible.

P7L3 – I suspect you meant to write "lambda > 307 nm" here?
P7L5 – Again, I suggest rewording to "an inverse relationship with the DEPTH OF DIFFERENTIAL SO2 ABSORPTION FEATURES..."

P7L14 - "... absorption minimum and STRAY light..."

P7L20 - "due TO the increasing..."

P7L20 and P8L18 – You mention "absorption non-linearity effects" here. In my opinion, this is a bit misleading because the DOAS model does not actually require the absorption to be linearly related to the dSCD. Instead, the optical depth (= logarithm of I/I0) is considered proportional to the SCD, as is described by the Beer-Lambert-Bouguer Law. The issue is actually more complex and has to do with the non-commutative nature of the convolution of absorption cross-sections and the application of the Beer-Lambert Law. Details can be found in Platt and Stutz (2008).

P7L34 – "fit error was > 1.1e16..." (remove two instances of "greater"/"greater than")

P8L11 – I did not understand this sentence. Can you clarify what you mean by "could be overestimated by the same windows for low concentration measurements."?

P8L13 – suggest adding "known" before "fit error for many windows".

P8L18 – Whether or not strong absorption effects related to the convolution contributed to the errors here might be assessed by comparing two synthetic spectra calculated for a known SCD: conv(exp(-sigmaHiRes\*SCD)) compared with exp(conv(sigmaHiRes)\*SCD)

P8L20 – "DEPTH OF features..."

P12L3 – "<10% less OR >10% more..."

P14Figure4 – Please provide units for the y axis.

P17Figure7 – It could be worth pointing out that using starting wavelengths of 304 vs 308.5 nm changes the results by an order of magnitude (!) for the base case.
References:

Lampel, J., Frieß, U., Platt, U., 2015. The impact of vibrational Raman scattering of air on DOAS measurements of atmospheric trace gases. Atmos. Meas. Tech. 8, 3767–3787. https://doi.org/10.5194/amt-8-3767-2015

Lampel, J., Wang, Y., Hilboll, A., Beirle, S., Sihler, H., Puk, J., Platt, U., Wagner, T., 2017. The tilt effect in DOAS observations. Atmos. Meas. Tech. 10, 4819–4831. https://doi.org/10.5194/amt-10-4819-2017

Langford, A.O., Schofield, R., Daniel, J.S., Portmann, R.W., Melamed, M.L., Miller, H.L., Dutton, E.G., Solomon, S., 2007. On the variability of the Ring effect in the near ultraviolet: understanding the role of aerosols and multiple scattering. Atmos. Chem. Phys. 7, 575–586. https://doi.org/10.5194/acp-7-575-2007

Platt, U., Stutz, J., 2008. Differential Optical Absorption Spectroscopy - Principles and Applications. Springer, Berlin, Heidelberg. https://doi.org/10.1007/978-3-540-75776-4

Stutz, J., Platt, U., 1996. Numerical analysis and estimation of the statistical error of differential optical absorption spectroscopy measurements with least-squares methods. Appl. Opt. 35, 6041–6053. https://doi.org/10.1364/AO.35.006041

Vogel, L., Sihler, H., Lampel, J., Wagner, T., Platt, U., 2013. Retrieval interval mapping: A tool to visualize the impact of the spectral retrieval range on differential optical absorption spectroscopy evaluations. Atmos. Meas. Tech. 6, 275–299. https://doi.org/10.5194/amt-6-275-2013

Vountas, M., Rozanov, V. V., & Burrows, J. P., 1998. Ring effect: impact of rotational Raman scattering on radiative transfer in Earht's atmosphere. J. Quant. Spectrosc. Radiat. Transfer, 60, 943-961, https://doi.org/10.1016/S0022-4073(97)00186-6

---

## Author Comment (AC2) · 13 May 2020

**Response to Christoph Kern - RC2**

**Interactive Comment from Review #2**

The author thanks Dr. Kern for reading and providing useful comments on the discussion paper.

**Specific Issues**

*Paragraph 1* It is my experience that imperfect representation of the differential ozone absorption in the 300 to 340 nm region in the DOAS model can lead to errors in retrieved SO2 column densities. These effects are not captured by the authors' experiments because they did not assess the effect of a varying solar zenith angle (SZA) and hence a change in the ozone SCD on their results. My worry is that these might affect the recommendations for wavelength fit window, possibly leading the authors to consider broadening the window to allow for better discrimination between SO2 and O3. To address this issue, the experiments could be repeated for a time early in the morning or late in the evening when the O3 SCD is changing rapidly with time…

- Response:

   Assessment of the impact of differential ozone absorption under varying solar zenith angle conditions are beyond the scope and has been added as a study limitation on Page 10 Lines 28-31. Recommendations that future work repeats the experiments under different solar geometry to address this issue are included on Page 10 Lines 31-32.

"A limitation of this study is the lack of measurements at high solar zenith angles (near dawn and dusk) when the SCDs of $O_3$ are larger and change rapidly with time. In such cases, fit accuracy may benefit from extending the upper limit of the fit window to allow better discrimination between the differential absorption features in the $O_3$ and $SO_2$ cross-sections."

*Paragraph 2* I recommend the authors review the literature with regards to state-of-the-art Ring correction, in particular focusing on an additional dependency of the Ring effect on wavelength (Vountas et al., 1998 Langford et al., 2007), the potential impact of vibrational Raman scattering (Lampel et al., 2015), and the effect of the broad-band shape of the solar spectrum as it reaches the Earth's surface (Lampel et al., 2017).

- Response to Dependency of the Ring on wavelength in Langford et al. (2007):

The inclusion of an adjustable parameter to the non-linear fitting steps of the DOAS analysis to compute the pseudo-absorption Ring cross-section in Langford et al. (2007) is mathematically non-trivial. It requires altering the DOASIS software algorithm or developing one's own algorithm, which is beyond the scope of this study. This study aims to provide information on some fitting parameters available in the widely used DOASIS software. The possibility of adding this correction to the analysis in a future study has been added on Page 10 lines 25-27.

"The DOAS analysis could also be expanded to include a correction to reduce impacts of wavelength dependence of the Ring effect in the near UV due to aerosol and multiple Rayleigh scattering as described in Langford et al. (2007)."

- Response to Potential Impact of vibrational Raman scattering in Lampel et al. (2015):

After becoming familiar with the potential impacts of vibrational Raman scattering (VRS) on DOAS fits and communicating with Dr. Johannes Lampel, it was determined that the VRS effect should be negligible in this deeper UV range. This is due to lack of strong Fraunhofer lines at wavelengths less than the fitting window to produce non-trivial $N_2$ and $O_2$ VRS components, the lack of light at these lower wavelengths present below the $O_3$ layer to be

scattered by VRS into the higher wavelengths of the fit window, and given the size of the fit residuals due to noise etc. In Lampel et al. (2015), the effect in the blue wavelength range was already small. Also, most of the added intensity due to VRS can be compensated by the offset polynomial in the absence of larger Fraunhofer or terrestrial absorption lines in the original intensity.

Response to the effect of the broad-band shape of the solar spectrum as it reaches the Earth's surface in Lampel et al. (2017):

Shift and squeeze terms were included in the fits not only for the absorption cross sections but also the FRS/Ring spectra, which should compensate for the tilt effect (Lampel, 2017). This information and a short description of the tilt effect is now included in the Methods section on Page 5 lines 10-19.

"The shift and squeeze terms were allowed for the fit components with the Ring spectrum terms linked to the FRS terms and the $O_3$ spectra terms linked to the $SO_2$ terms (shift limited to -0.2 to 0.2 nm). The shift and squeeze terms are included in DOAS analyses to compensate for wavelength shifts due to instrumental instabilities, such as temperature changes during measurements altering the pixel-to-wavelength calibration (Lampel et al., 2017; Stutz and Platt, 1996). In the case of the FRS, the shift and squeeze terms also compensate for the "tilt effect" that increases fit residuals by artificially shifting the spectral positions of Fraunhofer and molecular absorptions lines between the measurement and reference spectra that have different viewing elevation angles (Lampel et al., 2017). The tilt effect arises because atmospheric modification of the spectral structures in the spectrum occurs before convolution with the instrument slit function and are non-commutative but are applied in the reverse order by the analysis procedure (Lampel et al., 2017)."

*Paragraph 3* Finally, it's a bit unclear to me how representative the results are for MAX-DOAS instruments in general. The authors use an Ocean Optics USB2000 spectrometer for their measurements. This instrument is very common and therefore represents a good choice for such a study. However, it is my experience that these relatively economical instruments suffer from a relatively poor stray-light rejection. Therefore, it is no surprise that suppression of stray light using an optical filter and/or accounting for it in the DOAS fit improves the results of the retrieval. At the same time, these results and recommendations may not apply to other, higher quality instruments. It would be of great value if the authors were able to compare their results with those obtained using a higher-grade spectrometer. If this is not possible, the authors should consider explicitly narrowing the scope of their manuscript to reflect the fact that the experiments were all made using this one type of instrument. For example, the authors might include the terms 'low-cost' or 'miniature' in the title and mention the make and model of the spectrometer in the abstract and prominently throughout the manuscript.

- Response:

This is a good point, and the scope of the manuscript has been clarified by specifying that the measurements were conducted using a mini-MAX-DOAS instrument with a compact OceanOptics USB2000 spectrometer. The term 'miniature' (as 'mini') was added to the title. The spectrometer type and 'miniature' MAX-DOAS type was added to the abstract (Page 1 Line 10) and the Summary & Recommendations section (Page 9 lines 32-33). It is now noted that the OceanOptics USB2000 spectrograph is a relatively low-cost spectrometer in the Methods section (Page 4 line 15) and Summary & Recommendation Section (Page 11 Lines 16-18). It is also mentioned that these compact spectrometers tend to suffer from stray light impacts on Page 2 line 17.

Page 4 Line 15: "The MAX-DOAS instrument used **a relatively low-cost and commonly employed compact spectrometer**, an OceanOptics USB2000 spectrograph."

Page 11 lines 16-18 "Ultimately, the use of higher quality spectrometers with reduced stray light and improved spectral resolution for MAX-DOAS measurements is desirable, but a greater expense **compared to the low-cost spectrometer used in this study**."

**Minor Issues and Corrections**

**1)**    **P1L11** – I recommend rewording "the dSCDs also exhibited an inverse relationship with the DEPTH OF THE DIFFERENTIAL FEATURES IN THE SO2 absorption crosssection :"

Response: This sentence was reworded according to the recommendation (Page 1 lines 11).

**2)**    **P1L15** – ": : : dependence on the SO2 absorption features SUGGESTING THAT THE RADIANCE AT SHORTER WAVELENGTHS WAS increased by stray light: : :"

Response: This sentence was amended according to the recommendation (Page 1 line 17).

**3)**    **P1L18 and P7L25** – The uncertainty reported by the fit is not necessarily expected to be an accurate measure of the errors of the results. This is discussed by Stutz and Platt (1996). Please incorporate this information into your discussion.

Response: This is a good point and the information was incorporated into the discussion on Page 9 Lines 7-14 . A set of figures showing the proportion of the fit uncertainty to the fit error for the $2^\circ$ spectra of both cell concentrations was added to the supplemental (Fig. S2 on Supplement Page 2) for the reader to be able to interpret the percentage degree of deviation between the values. This figure is referred to in the manuscript on Page 9 on lines 22, 25 and 28.  The percentage deviation is also referred to in the abstract on Page 1 line 20.

Page 9 Lines 7-14: "While the fit uncertainty reported by the retrieval is commonly used as the error on the retrieved dSCD, this uncertainty is not always expected to accurately represent the true error due to factors including assumptions about the independence of errors,  the presence of noise in the spectrum and structures in the fit residual (Stutz and Platt, 1996). Tests of modelled spectra with noise added found that when noise becomes large, the true errors of the retrieved trace-gas coefficient were >10% larger than the retrieved error and the difference was proportional to the noise level. Also, the inclusion of random residual structures in the spectra caused the fit uncertainty to underestimate the true error by a factor of 3 (Stutz and Platt, 1996). It is useful to examine which fitting windows exhibited the greatest difference between the fit uncertainty and error, shown in the right columns of Figs. 9 and 10."

**4)**    **P2L6** – I suggest adding "typically" before "uses the SO2 B band: : :"

Response: "Commonly" was added (Page 2 Line 7).

**5)**    **P2L15** – In my experience, it can be beneficial to include wavelengths with weak or negligible SO2 absorption in the fit if the extended wavelength range allows for better discrimination between SO2 and other aspects of the DOAS model (Ozone absorption, Ring effect).

Response: Agreed, the reviewer makes a good point. This clarification was included as "Inclusion of upper wavelengths with weak $SO_2$ in the fit can improve the fit results by allowing a better distinction between $SO_2$ absorption features and other fit components (e.g., $O_3$ absorption features, Ring spectrum)." (Page 2 line 26-29).

**6)**    **P3L6** – ": : : may NOT BE IDEAL for smaller: : :"

Response: Amended accordingly (Page 3 Line 32).

**7)**    **P3L23** – please remove the superscript formatting on the "C".

Response: Removed on Page 4 Line 19.

**8)**      **P3L30** – Please clarify how the 87 ppb were arrived at. I think this is assuming that the 30 degree spectra were evaluated relative to a reference recorded looking towards the zenith through the same atmospheric boundary layer, correct?

Response: The calculations had an error and have now been updated to the correct values of 41 and 4 ppb on Page 4 Line 26. These values were determined by calculating the $SO_2$ VCD using Eq. (2) in Davis et al. (2019) since the FRS was measured without the cell and should have negligible $SO_2$, similar to the Davis et al. (2019) mobile-MAX-DOAS measurements using Eq. (2) where the measurement spectrum observed within a polluted mass but the FRS observed in a clean region within a short time period. The method was clarified and reference to this equation was added on Page 4 Lines 27-28. The mixing ratio was then calculated assuming a boundary layer height of 1000 m contained all the $SO_2$ in the VCD and the density of air given standard temperature and pressure conditions.

**9)**      **P3L31** – Please include the time at which the measurements were made. This will allow the readers to deduce the solar zenith angle. How far apart in time where the 2-degree and 90-degree measurements made?

Response: The measurements were conducted on September 23 between 12:53 and 13:26 (local time). The time between the $2^o$ and $90^o$ measurements in the same sequence (both containing the $SO_2$ cell) was less than 13 minutes. The zenith spectra without a cell used as the FRS were obtained less than 35 minutes after the respective cell measurements. This information was added on page 3 lines 29-33.

**10)**     **P4L2** – In the future, you might consider using a Hoya U330 filter. I believe this has better rejection of NIR radiation which can cause stray light in the spectrometer.

Response: Thanks for the suggestion. We will try this filter in future.

**11)**     **P5L23** – Wavelengths longer than 324 nm are commonly used in volcanic gas measurements. While they admittedly often encounter larger SO2 SCDs, it's not clear to me that these wavelengths can be discounted across the board for MAX-DOAS measurements simply based on the increased DOF. See comment above on improved discrimination between SO2, ozone and Ring features.

Response: Agreed. The sentence referring to DOF was removed (Page 7 lines 2-3). The potential benefit of extending the upper limit to longer wavelengths has been added to Introduction on Page 2 lines 26-29.

"However, an overly narrow fit window can lead to cross-correlation between the reference absorption cross-sections (Vogel et al., 2013). Inclusion of upper wavelengths with weak $SO_2$ in the fit can improve the fit results by allowing a better distinction between $SO_2$ absorption features and other fit components (e.g., $O_3$ absorption features, Ring spectrum)."

**12)**     **P5L29** – It's not clear to me how increased Rayleigh scattering due to higher air pressure would preferentially remove shorter UV wavelengths. Is there a citation that you could provide for this? Couldn't one just as easily argue that increased Rayleigh scattering of previously unscattered sunlight would increase the measured radiance at shorter wavelengths? I suspect that another effect or combination of effects is responsible, possibly having to do with radiation being removed from the atmospheric half-sphere by absorption on the ground. In this case, it might be the wavelength dependence of the ground albedo that is responsible.

Response: Since the cause(s) are uncertain for the relatively smaller UV signal in lower compared to higher elevation angle measured spectra this sentence was removed, it being not essential to the manuscript (Page 6 lines 7-9).

**13)**     **P7L3** – I suspect you meant to write "lambda > 307 nm" here?

Response: Correct, thanks for catching that. Corrected on Page 8 line 16.

**14)**     **P7L5** – Again, I suggest rewording to "an inverse relationship with the DEPTH OF DIFFERENTIAL SO2 ABSORPTION FEATURES: : :"

Response: Reworded accordingly on Page 8 Line 18.

**15)**     **P7L14** – ": : : absorption minimum and STRAY light: : :"

Response: Typo corrected on Page 9 line 29.

**16)**     **P7L20** – "due TO the increasing: : :"

Response: Typo corrected on Page 9 line 1.

**17)**     **P7L20 and P8L18** – You mention "absorption non-linearity effects" here. In my opinion, this is a bit misleading because the DOAS model does not actually require the absorption to be linearly related to the dSCD. Instead, the optical depth (= logarithm of I/I0) is considered proportional to the SCD, as is described by the Beer-Lambert-Bouguer Law. The issue is actually more complex and has to do with the non-commutative nature of the convolution of absorption cross-sections and the application of the Beer- Lambert Law. Details can be found in Platt and Stutz (2008).

Response: Agreed, the comments on "non-linearity" effects were unclear. The non-linearity phenomenon is now explained in greater detail in the Introduction on Page 2 line 33 to Page 3 line 13. The effect also was better clarified in the results and summary sections on Page 6 lines 25-31 and Page 10 line 11-13, respectively.

Page 2 line 33 – Page 3 line 13: "A further complication is that for measurements of very large column densities of $SO_2$ (e.g., from volcanic studies), the optimal wavelength window may be present at higher wavelengths where $SO_2$ absorption features are weaker (Bobrowski et al., 2010). High optical densities below 320 nm from large column densities can cause non-linearities in the relationship between the column density and measured optical density in the fit. This phenomenon occurs for large (actual) optical densities if the cross-section in the fit was not recorded with the same spectrometer as the measurements, which is common, and the instrument's spectral resolution is too low to completely resolve the absorption bands (Kern, 2009; Platt and Stutz, 2008). Large column densities of $SO_2$ result in optical densities in the B band that can exceed unity, violating the assumption in the standard DOAS retrieval of weak absorption with optical depths of less than ~0.1 (Bobrowski et al., 2010; Bobrowski and Platt, 2007; Fickel and Delgado Granados, 2017; Kern, 2009; Platt and Stutz, 2008). Compact spectrometers typically have an insufficient spectral resolution for the optical density of the $SO_2$ absorption bands to be proportional after convolution for large column densities (Bobrowski et al., 2010; Platt and Stutz, 2008). Consequently, the true column density can be underestimated because the differential absorption line depths from the standard DOAS convolution approximation can be greater than mathematically correct convolution (Bobrowski et al., 2010; Kern, 2009; Yang et al., 2007). Underestimation has been shown to increase with decreasing wavelength from 320-300 nm and increasing column density of $SO_2$ (Kern, 2009)."

**18)**     P7L34 – "fit error was > 1.1e16: : :" (remove two instances of "greater"/"greater than")

Response: First instance of "greater" removed on Page 9 line 25.

**19)**     **P8L11** – I did not understand this sentence. Can you clarify what you mean by "could be overestimated by the same windows for low concentration measurements."?

Response: The sentence is referring to the "direction" of the error in $SO_2$ dSCD being inconsistent for windows with lower limit <307 nm between the higher and lower concentration cells. That is, the higher cell retrievals from these windows generally overestimated dSCD but the lower cell retrievals often underestimated. The sentence was edited and is, hopefully, now clear on Page 10 Lines 4-6.

**20)**      P8L13 – suggest adding "known" before "fit error for many windows".

Response: "Known" added on Page 10 line 7.

**21)      P8L18** – Whether or not strong absorption effects related to the convolution contributed to the errors here might be assessed by comparing two synthetic spectra calculated for a known SCD: conv(exp(-sigmaHiRes*SCD)) compared with exp(-conv(sigmaHiRes)*SCD)

Response: This sentence has been removed as now superfluous based on the greater clarification and discussion of non-linearity effects in Section 3.1 on Page 6 lines 25-31. This includes referring the reader a figure in Kern (2009) showing the percentage underestimation of the $SO_2$ column density under varying actual column densities that include the levels tested here using a similar spectral resolution (Page 6 lines 29-31).

**22)**      P8L20 – "DEPTH OF features: : :"

Response: Amended accordingly on Page 10 line 17.

**23)**      P12L3 – "<10% less OR >10% more: : :"

Response: Corrected on Page 15 line 1.

**24)**      P14Figure4 – Please provide units for the y axis.

Response: Units added to Y-axis of Figure 4 on Page 16.

**25)      P17Figure7** – It could be worth pointing out that using starting wavelengths of 304 vs 308.5 nm changes the results by an order of magnitude (!) for the base case.

Response: This observation has been added on Page 8 lines 22-24.

**Citations Mentioned**

Bobrowski, N., Kern, C., Platt, U., Hoermann, C. and Wagner, T.: Novel SO2 spectral evaluation scheme using the 360-390 nm wavelength range, Atmospheric Meas. Tech., 3(4), 879–891, doi:10.5194/amt-3-879-2010, 2010.

Davis, Z. Y. W., Baray, S., McLinden, C. A., Khanbabakhani, A., Fujs, W., Csukat, C., Debosz, J. and McLaren, R.: Estimation of NOx and SO2 emissions from Sarnia, Ontario, using a mobile MAX-DOAS (Multi-AXis Differential Optical Absorption Spectroscopy) and a NOx analyzer, Atmospheric Chem. Phys., 19(22), 13871–13889, doi:10.5194/acp-19-13871-2019, 2019.

Fickel, M. and Delgado Granados, H.: On the use of different spectral windows in DOAS evaluations: Effects on the estimation of SO2 emission rate and mixing ratios during strong emission of Popocatepetl volcano, Chem. Geol., 462, 67–73, doi:10.1016/j.chemgeo.2017.05.001, 2017.

Kern, C.: Spectroscopic measurements of volcanic gas emissions in the ultra-violet wavelength region, Ph.D. Thesis, University of Heidelberg, Germany, 318 pp., https://doi.org/10.11588/heidok.00009574, 2009.

Lampel, J., Frieß, U. and Platt, U.: The impact of vibrational Raman scattering of air on DOAS measurements of atmospheric trace gases, Atmospheric Meas. Tech., 8(9), 3767–3787, doi:https://doi.org/10.5194/amt-8-3767-2015, 2015.

Lampel, J., Wang, Y., Hilboll, A., Beirle, S., Sihler, H., Puķīte, J., Platt, U. and Wagner, T.: The tilt effect in DOAS observations, Atmospheric Meas. Tech., 10(12), 4819–4831, doi:https://doi.org/10.5194/amt-10-4819-2017, 2017.

Langford, A. O., Schofield, R., Daniel, J. S., Portmann, R. W., Melamed, M. L., Miller, H. L., Dutton, E. G. and Solomon, S.: On the variability of the Ring effect in the near ultraviolet: understanding the role of aerosols and multiple scattering, Atmospheric Chem. Phys., 7(3), 575–586, doi:https://doi.org/10.5194/acp-7-575-2007, 2007.

Platt, U. and Stutz, J.: Differential optical absorption spectroscopy : principles and applications, Springer Verlag, Berlin., 2008.

Pukite, J., Kuehl, S., Deutschmann, T., Platt, U. and Wagner, T.: Extending differential optical absorption spectroscopy for limb measurements in the UV, Atmospheric Meas. Tech., 3(3), 631–653, doi:10.5194/amt-3-631-2010, 2010.

Theys, N., De Smedt, I., van Gent, J., Danckaert, T., Wang, T., Hendrick, F., Stavrakou, T., Bauduin, S., Clarisse, L., Li, C., Krotkov, N., Yu, H., Brenot, H. and Van Roozendael, M.: Sulfur dioxide vertical column DOAS retrievals from the Ozone Monitoring Instrument: Global observations and comparison to ground-based and satellite data, J. Geophys. Res.-Atmospheres, 120(6), 2470–2491, doi:10.1002/2014JD022657, 2015.

Theys, N., De Smedt, I., Yu, H., Danckaert, T., van Gent, J., Hoermann, C., Wagner, T., Hedelt, P., Bauer, H., Romahn, F., Pedergnana, M., Loyola, D. and Van Roozendael, M.: Sulfur dioxide retrievals from TROPOMI onboard Sentinel-5 Precursor: algorithm theoretical basis, Atmospheric Meas. Tech., 10(1), 119–153, doi:10.5194/amt-10-119-2017, 2017.

Theys, N., Hedelt, P., De Smedt, I., Lerot, C., Yu, H., Vlietinck, J., Pedergnana, M., Arellano, S., Galle, B., Fernandez, D., Carlito, C. J. M., Barrington, C., Taisne, B., Delgado-Granados, H., Loyola, D. and Van Roozendael, M.: Global monitoring of volcanic SO 2 degassing with unprecedented resolution from TROPOMI onboard Sentinel-5 Precursor, Sci. Rep., 9(1), 1–10, doi:10.1038/s41598-019-39279-y, 2019.

---

## Author Comment (AC1)

**Recommendations for Spectral Fitting of SO2 from mini-MAX-DOAS Measurements**

**Zoë Y. W. Davis and Robert McLaren**

**Response to RC1**

**Interactive Comment from Review #1**

The author thanks referee #1 for reading and providing useful comments on the discussion paper.

**General Considerations**

1) How would you use the recommendations discussed in the article with satellite comparisons?

Response: The recommendations are not applicable for satellite measurements since the spectrometers used in satellites are typically much higher quality, and the radiative transfer can be much different. The recommendations in the current study are intended to be representative of low-cost, compact spectrometers such as the OceanOptics USB2000, which is commonly used in mini-MAX-DOAS measurements. For comparison, the TROPOMI satellite spectrometer has a signal-to-noise ratio of 1000:1 and spectral resolution of 0.54 nm in the SO2 fitting region compared to 250:1 and 0.72 nm for this MAX-DOAS spectrometer. The scope of the recommendations has been greater clarified by including the term 'miniature' (as 'mini') in the title. The spectrometer type and 'miniature' MAX-DOAS type was also added to the abstract (Page 1 Line 10) and the Summary & Recommendations section (Page 9 lines 32-33). It is also now noted that the OceanOptics USB2000 spectrograph is a relatively low-cost spectrometer in the Methods section (Page 4 line 15) and Summary & Recommendation Section (Page 11 Line 16-18).

Page 4 Line 15: "The MAX-DOAS instrument used **a relatively low-cost and commonly employed compact spectrometer**, an OceanOptics USB2000 spectrograph."

If this comment was referring to the utility of comparing MAX-DOAS and satellite measurements of SO2: MAX-DOAS measurements are well suited to help validate satellite retrievals as satellite measurements gain increasingly better horizontal pixel resolution that becomes closer to the spatial resolution of MAX-DOAS instruments (i.e., a few kilometres). For example, MAX-DOAS VCDs could provide important validation for satellite retrievals of trace-gases, which depend on modelled a-priori profiles of the gases that may not resolve small areas.

2) Have you tried the 315-325 nm window fit?

Response: Yes, the results of the 315-325 nm window fit are shown in Figs. 2 and 5 for the higher and lower concentration cells, respectively. This window overestimates the true SO2 dSCD by >10% and >50% for the higher and lower concentration cells, respectively, for all fit scenarios and both the 2° and 30° measurement spectra (Figs. 2 and 5).

3) Have you thought in a comparison, using the window fit recommend, with satellite? OMI? TROPOMI?

Response: The recommended fit window is not intended for use with satellite measurements because spectrometers in satellites are typically much higher quality with higher resolution and reduced stray light (see response to General Consideration Comment #1). Also, satellite measurements can have additional fitting issues affecting the optimal fitting window, and they often observe much greater SCDs (e.g., over volcanoes) than in urban areas.

The recommended window is not appropriate for satellites since satellite retrievals (e.g., GOME, OMI) use the 310-326 nm range. This range is used because increased Rayleigh scattering and  $O_3$  absorption results in a small  $SO_2$  signal relative to  $O_3$  absorption in the spectrum at lower wavelengths (Theys et al., 2017). For example, TROPOMI retrievals use the 312-326 nm window as a baseline for SCDs  $\leq 4x10^{17}$  molec. cm-2 (15 DU) (Theys et al., 2017). Changing the window to 310.5-326 nm to include an additional SO2 band reduced sensitivity to lower troposphere absorption by SO2 and only a small improvement in data scatter from OMI satellite measurements of SO2, attributed to decreased intensity and signal-to-noise ratio in the short UV due to strong absorption by O3 (Theys et al., 2015). Additional fitting issues specific to satellite retrievals of SO2 compared to MAX-DOAS include much longer light path lengths requiring added corrections for wavelength dependence of the light path and absorption by trace-gases, such as O3, (Pukite et al. 2010) that are used in the TROPOMI retrievals (Theys et al., 2017).

Note that for TROPOMI retrievals of SO2 SCDs >4x1017 molec. cm-2 (15 DU), an alternate window of 325-335 nm is used. If the first alternate window retrieves an SCD >250 DU, the second alternate fitting window of 360-390 nm is used to avoid signal saturation (Theys et al., 2017; Theys et al., 2019). The need for fitting windows at higher wavelengths for very large SO2 column densities is more fully discussed in the Introduction (Page 2 lines 33 to Page 3 line 13). Therefore, using the recommended fit window for satellite measurements would likely result in underestimated SO2 for moderate to large SCDs values.

Page 2 line 33 – Page 3 line 13: "A further complication is that for measurements of very large column densities of  $SO_2$  (e.g., from volcanic studies), the optimal wavelength window may be present at higher wavelengths where  $SO_2$ absorption features are weaker (Bobrowski et al., 2010). High optical densities below 320 nm from large column densities can cause non-linearities in the relationship between the column density and measured optical density in the fit. This phenomenon occurs for large (actual) optical densities if the cross-section in the fit was not recorded with the same spectrometer as the measurements, which is common, and the instrument's spectral resolution is too low to completely resolve the absorption bands (Kern, 2009; Platt and Stutz, 2008). Large column densities of SO2 result in optical densities in the B band that can exceed unity, violating the assumption in the standard DOAS retrieval of weak absorption with optical depths of less than ~0.1 (Bobrowski et al., 2010; Bobrowski and Platt, 2007; Fickel and Delgado Granados, 2017; Kern, 2009; Platt and Stutz, 2008). Compact spectrometers typically have an insufficient spectral resolution for the optical density of the SO2 absorption bands to be proportional after convolution for large column densities (Bobrowski et al., 2010; Platt and Stutz, 2008). Consequently, the true column density can be underestimated because the differential absorption line depths from the standard DOAS convolution approximation can be greater than mathematically correct convolution (Bobrowski et al., 2010; Kern, 2009; Yang et al., 2007). Underestimation has been shown to increase with decreasing wavelength from 320-300 nm and increasing column density of SO2 (Kern, 2009)."

**4) Is it necessary to correct the ring effect if active DOAS is used?**

Response: It is not necessary to correct for the Ring effect if using active DOAS. For cavity enhanced active DOAS (CE-DOAS), the Ring effect is not an issue because the light source is artificial, and the light path is inside a closed cavity. For long path DOAS (LP-DOAS), the instrumental set-up (e.g., path length and retroreflector properties) ensures that the lamp signal is strong enough for the contribution of scattered light to the measured spectrum to be trivial enough not to require correction.

**5)** It would be very useful to include in the article measurement of calibration cells with high SO2 concentrations ( $\sim 10^{19}$ ), applying the recommendations presented in the article and discussing the results.

Response: Measurements of such high concentrations are outside of the scope of this study since the aim was to represent a range of SO2 expected in urban settings, as stated on Page 4 Line 9. Other studies have addressed retrieving SO2 using DOAS at such higher column densities, including Bobrowski et al. (2010) and Fickel and Delgado Granados (2017), and the readers are referred to these works in the Summary and Recommendations section (Page 9 Line 34 to Page 10 Line 1). The Bobrowski et al. (2010) may be of interest here as it provides examples of using different fitting windows for SO2 from both MAX-DOAS and GOME-2 satellite measurements.

Future MAX-DOAS studies could repeat the current study with such higher concentrations (3 cells) for applications where both anthropogenic ambient/megacity and volcano emissions are to be observed.

**Specific Consideration**

1/ Introduction Line 34: Rix et al., 2012, spectral range fit between 315 and 326 nm.

Response: Upper range value corrected on Page 2 Line 13.

2/ Introduction Line 43: verify reference "Wang and Christopher, 2003", is it correct?

Response: The reference was incorrect, thanks for catching the mistake. The reference was replaced by the correct reference, Tan et al. (2018) on Page 3 Line 25.

3/ Methods Line 55: Instrument resolution (FWHM) are calculated or is the information provided by the company (in this case, Ocean Optics)? The company provides an average resolution of the entire spectral range of the instrument and not in the specific absorption range of a given compound (e.g. SO2, NO2, etc..).

Response: The FWHM was measured using a Hg lamp and Gauss fitting the Hg emission line at 334 nm. The value of 0.7 nm on Page 4 Line 18. is the measured value, rather than the provided value.

4/ Methods Line 58: "using the DOASIS software package". Why using DOASIS? In my opinion, QDOAS is currently the best and most widely used software for the fitting of contaminating compounds in the atmosphere with ground-based techniques.

Response: DOASIS is commonly used software for MAX-DOAS applications and has been used for all DOAS measurements in our research group. For future measurements, we can try the recommended QDOAS software.

5/ Line 61-62: Be careful to use high and low concentration referring to the calibration cells. I disagree in considering a concentration of 1e17 as "high". This is relative. What are you talking about? Contaminated areas? industrial areas? volcanic areas? Megacities? The SO2 concentrations of 1e17 molec/cm2 (occasionally 1e18) are "high" considering anthropogenic contaminated areas, such as megacities or/and industrial areas. Volcanic SO2 concentrations can reach 1e19 molec/cm2. Using instruments such as UV camera, very concentrated calibration cells are needed (1e19 molec/cm2), to be able to calibrate the instrument for SO2 concentrated plumes. I suggest specifying the "high concentrations" in the atmosphere of polluted areas, like megacities.

Response: This is a good point. For greater clarity, the "high" and "low" concentration cell measurements are now referred to as "higher" and "lower" throughout the manuscript and supplemental. It is specified that the calibration cells are intended to be representative of polluted urban conditions in the introduction on Page 4 Line 9. This point and the presence of greater SCDs in volcanic or major industrial areas are included in the Summary & Recommendations section on Page 10 Lines 33-34. This includes references to studies relevant to fitting such SO2 greater SCDs.

Page 10 Lines 33-34 "Greater SO2 column densities (> $1x10^{18}$  molec. cm-2) can be observed in volcanic areas and close to major industrial sources; discussions of retrieving such greater SO2 column densities can be found in Bobrowski et al. (2010) and Fickel and Delgado Granados (2017)."

**6**/ **Introduction Line 63:** "These SCDs would be equivalent to an air mass with SO2 mixing ratios of 87 and 8.7 ppb: : :". Is the calculation correct? How did you calculate these values?

Response: The calculations had an error and have now been corrected to the values of 41 and 4 ppb. These values were determined by calculating the SO2 VCD using Eq. (2) in Davis et al. (2019) since the FRS was measured without the cell and should have negligible SO2, similar to the conditions of the Davis et al. (2019) mobile-MAX-

DOAS measurements using Eq. (2) where the measurement spectrum observed within a polluted mass, but the FRS observed in a clean region within a short time period. The reference to this equation was added on Page 4 Lines 25-28. The mixing ratio was then calculated, assuming a boundary layer height of 1000 m contained all the  $SO_2$  in the VCD and the density of air given standard temperature and pressure conditions.

**Final Consideration**

Personally, I tried to apply the recommendations, discussed in the article, on the best SO2 fitting windows. I compared, using MAX-DOAS and Direct Sun techniques, the 307.5-319 nm window fit with other windows fit used in the literature. The 307.5-319 nm window fit work well in anthropogenic contaminated areas, I was able to recover more spectra and obtaining a lower RMS than other windows fit. On the other hand, the results obtained by testing the recommended window fit with very concentrated calibration cells (\_1e19), greatly underestimate the real concentration inside the cells.

Response: See the responses to comments General Considerations #5 and Specific Considerations #5.